# FAIL: Flow Matching Adversarial Imitation Learning for Image Generation

Yeyao Ma [1]  Chen Li [2]  Xiaosong Zhang [3]  Han Hu [3]  Weidi Xie [1]

## Abstract

Post-training of flow matching models—aligning the output distribution with a high-quality target—is mathematically equivalent to imitation learning. While Supervised Fine-Tuning mimics expert demonstrations effectively, it cannot correct policy drift in unseen states. Preference optimization methods address this but require costly preference pairs or reward modeling. We propose Flow Matching Adversarial Imitation Learning (FAIL), which minimizes policy-expert divergence through adversarial training without explicit rewards or pairwise comparisons. We derive two algorithms: FAIL-PD exploits differentiable ODE solvers for low-variance pathwise gradients, while FAIL-PG provides a black-box alternative for discrete or computationally constrained settings. Fine-tuning FLUX with only 13,000 demonstrations from Nano Banana pro, FAIL achieves competitive performance on prompt following and aesthetic benchmarks. Furthermore, the framework generalizes effectively to discrete image and video generation, and functions as a robust regularizer to mitigate reward hacking in reward-based optimization. Code and data are available at https://github.com/HansPolo113/FAIL

## 1. Introduction

Flow Matching (Lipman et al., 2022; Liu et al., 2022) and Diffusion (Ho et al., 2020; Song et al., 2020) models have emerged as the dominant paradigms for image generation, enabling the synthesis of high-fidelity, diverse visual content. The training lifecycle of these models is generally bifurcated into two distinct stages: pretraining and post-training. Pretraining focuses on broad knowledge acquisition and mode coverage, allowing the model to learn the underlying physics

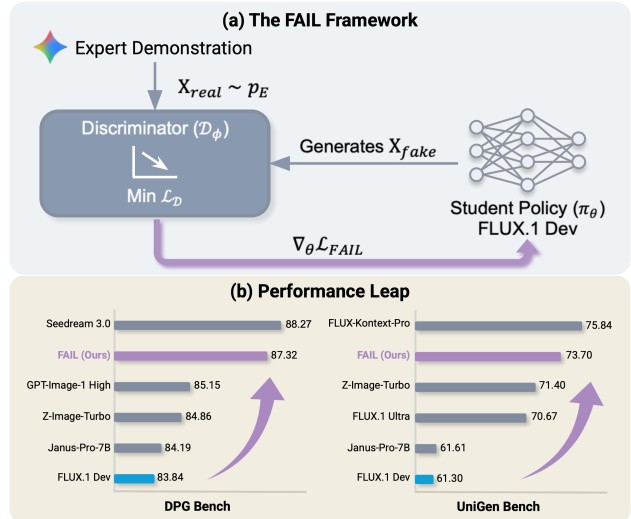

*Figure 1.* (a) We propose FAIL, an adversarial imitation learning framework for flow matching model. (b) With 13K limited data, FAIL significantly improved performance of FLUX baseline.

of visual data. Conversely, post-training aims to align the model with a specific, high-quality "preferred distribution", ensuring the generation of aesthetically pleasing and prompt-faithful images. Currently, two primary paradigms govern this alignment phase: Supervised Fine-Tuning (SFT) and Preference Optimization (*e.g.*, RLHF (Ouyang et al., 2022), DPO (Rafailov et al., 2023)). SFT serves as the foundational step, guiding the model to approximate the preferred distribution through direct demonstration. Preference Optimization builds upon this by refining the model's outputs based on comparative feedback or scalar rewards, pushing the policy toward regions of higher utility. Together, these methods enable the backbone of modern generative alignment strategies.

We propose that the objective of post-training—aligning the model's output distribution with an ideal target distribution—is mathematically equivalent to Imitation Learning (IL). Under this framework, SFT corresponds to Behavioral Cloning (BC) (Pomerleau, 1991a). While data-efficient, SFT suffers from distribution shift problem, where the policy drifts when encountering states outside the expert demonstrations, leading to compounding errors. Preference Optimization methods (*e.g.*, RLHF, DPO) effectively address this drift by utilizing comparative feedback. However,

[1]Shanghai Jiao Tong University [2]Xi'an Jiaotong University [3]Tencent. Correspondence to: Weidi Xie <weidi@sjtu.edu.cn>.

*Proceedings of the $43^{rd}$ International Conference on Machine Learning*, Seoul, South Korea. PMLR 306, 2026. Copyright 2026 by the author(s).

the widespread adoption of these methods is constrained by their reliance on high-quality preference data and the complexity of training accurate reward models.

Adversarial Imitation Learning (AIL) (Ho & Ermon, 2016; Fu et al., 2017a) emerges as a promising alternative that addresses these limitations while maintaining robust alignment capabilities. Rather than relying on explicit reward modeling or preference pairs, AIL minimizes the Jensen-Shannon divergence between the policy and expert distributions via a minimax game (Goodfellow et al., 2020). By employing a discriminator to distinguish between generated and expert samples, AIL provides a dense, learned training signal that guides the policy toward the target distribution.

In this work, we introduce **Flow Matching Adversarial Imitation Learning (FAIL)**, a framework that adapts adversarial imitation to continuous-time generative dynamics. We instantiate FAIL via two distinct gradient estimation mechanisms. The primary approach, **FAIL-PD (Pathwise Derivative)**, is a white-box method that leverages the differentiable nature of the ODE solver in Flow Matching. By backpropagating the discriminator's guidance directly through the denoising steps to the policy parameters, FAIL-PD provides a lower-variance, richer learning signal that results in superior final performance. As a supplementary approach, we introduce **FAIL-PG (Policy Gradient)**, a black-box method utilizing the Score Function Estimator. FAIL-PG is essential for scenarios where direct backpropagation is intractable—such as autoregressive generation with discrete tokens—or when computing gradients through a massive discriminator is computationally prohibitive.

We empirically validate our approach by fine-tuning the FLUX (Labs, 2024) model using synthetic images from Gemini 3 Pro (Google, 2025) as the expert dataset. Crucially, our experimental setup restricts the data to a single expert image per prompt to simulate real-world scenarios where collecting preference pairs is impractical. Despite this challenging regime and a modest dataset size of 13,000 samples, FAIL delivers substantial performance gains. On UniGen-Bench, FAIL-PD improves the base model from 61.61 to 73.70, surpassing models like FLUX.1 Ultra. Similarly, on DPG-Bench, it boosts performance to 87.32. Furthermore, we demonstrate that FAIL integrates seamlessly into standard Reinforcement Learning frameworks, effectively stabilizing training and preventing reward hacking.

Our contributions can be summarized as follows:

- We formulate the post-training of Flow Matching models as an Adversarial Imitation Learning problem, proposing the **FAIL** framework to bridge the gap between Supervised Fine-Tuning and Reinforcement Learning.

- We derive two practical algorithms: **FAIL-PD**, which utilizes pathwise derivatives for high-performance align-

ment, and **FAIL-PG**, a flexible alternative for discrete or computationally constrained scenarios.

- We demonstrate the efficacy of FAIL on the FLUX, showing that it achieves competitive performance with minimal, single-sample data and can serve as a stabilizer for other RL-based fine-tuning methods.

## 2. Method

We introduce **Flow Matching Adversarial Imitation Learning (FAIL)**, a framework for learning continuous normalizing flows from expert demonstrations without explicit reward functions. We formulate the problem as a minimax game between a generator policy $\pi_\theta$ and a discriminator $D_\omega$. The objective is to find a policy that generates samples indistinguishable from the expert distribution $p_E(x)$, minimizing the divergence:

$$\min_\theta \max_\omega \mathbb{E}_{x \sim p_E}[\log D_\omega(x)] + \mathbb{E}_{x \sim \pi_\theta}[\log(1 - D_\omega(x))]. \quad (1)$$

To solve this optimization, we propose two strategies for the policy update: FAIL-PD, which exploits the differentiability of the ODE solver to compute pathwise derivatives, and FAIL-PG, a black-box policy gradient approach leveraging a tractable likelihood ratio estimator.

### 2.1. FAIL-PD: Pathwise Derivative Optimization

Unlike standard Reinforcement Learning (RL) settings where environment dynamics are unknown and non-differentiable, the "environment" in Flow Matching is a deterministic Ordinary Differential Equation (ODE) solver. This structure allows us to backpropagate the discriminator's learning signal directly through the generation process to update the vector field.

In FAIL-PD, the policy generates a sample $x_0$ by integrating the vector field $v_\theta$ starting from noise $\epsilon \sim \mathcal{N}(0, I)$. The discriminator $D_\omega$ evaluates $x_0$, acting as a differentiable reward function. Since the numerical integration steps (*e.g.*, Euler or RK4) are differentiable operations, we can compute the exact gradient of the discriminator's output with respect to the flow parameters $\theta$ via the chain rule.

To reduce the computational cost of backpropagating through the entire trajectory, we employ a single-step denoising approximation. Instead of unrolling the full ODE chain for every gradient step, we sample a random timestep $t \sim \mathcal{U}(0, 1)$, inject noise to obtain $x_t$, and perform a single denoising step. We estimate the clean data $x_0'$ via the following update rule:

$$x_0' = \frac{x_t + \Delta t \cdot v_\theta(x_t, t) - (t + \Delta t)\epsilon}{1 - (t + \Delta t)}, \quad (2)$$

where $x_t = (1 - t)x_0 + t\epsilon$ is the interpolated state and $\Delta t$ corresponds to the step size used during inference. This

**Algorithm 1** FAIL-PD: Flow Matching Adversarial Imitation Learning with Pathwise Derivative

1: **Input:** Expert images $x_E \sim \pi_E$, initial policy $\pi_\theta$ and discriminator $D_\omega$ with parameters $\theta_0, \omega_0$
2: **for** $i = 0, 1, 2, \ldots$ **do**
3:      Sample a batch of images $\{x_E^{(i)}\}$ from $x_E$
4:      For each prompt of $\{x_E^{(i)}\}$, sample $G$ image noise pairs $(x_0^j, \epsilon^j)$ where $\epsilon \sim \mathcal{N}(0, I)$
5:      Update discriminator from $\omega_i$ to $\omega_{i+1}$ with gradient

$$\hat{\mathbb{E}}_{x_E}[\nabla_\omega \log \sigma(D_\omega(x_E))] + \hat{\mathbb{E}}_{x_0}[\nabla_\omega \log(1 - \sigma(D_\omega(x_0)))]$$

6:      For each $(x_0, \epsilon)$, sample $t \sim \mathcal{U}(0, 1)$, obtain $x_0'$ via single step denoising approximation using Equation 2
7:      Update policy from $\theta_i$ to $\theta_{i+1}$ with gradient

$$\hat{\mathbb{E}}_{x_0,t,x_1} \left[ -\nabla_\theta \log \sigma(D_{\omega_{i+1}}(x_0')) \right]$$

8: **end for**

---

approximation assumes local linearity of the vector field, allowing efficient updates to $\pi_\theta$ at arbitrary timesteps without full trajectory generation. See Algorithm 1 for detail.

**Theoretical Connection.** FAIL-PD can be analyzed through the lens of the Deterministic Policy Gradient (DPG) (Silver et al., 2014) framework. In standard RL, DPG relies on a learned critic $Q(s, a)$ to estimate gradients because the dynamics are unknown. Here, the ODE solver represents known, differentiable dynamics, and the discriminator acts as the reward function. By differentiating through the solver, we implicitly compute the exact action-value gradient without a separate critic.

Consequently, FAIL-PD can be viewed as a low-variance, unbiased limit of DDPG (Lillicrap et al., 2015) applied to Imitation Learning, inheriting the local convergence properties of Generative Adversarial Networks (GANs).

## 2.2. FAIL-PG: Policy Gradient Optimization

While FAIL-PD is efficient, computing gradient can be computationally prohibitive for large-scale discriminators or intractable for discrete flow formulations. To address this, we present FAIL-PG, a policy gradient alternative that simplifies gradient estimation via a tractable expectation.

Instead of differentiating through the dynamics, FAIL-PG treats the discriminator output as a scalar reward, $r(x) = -\log(1 - \sigma(D_\omega(x)))$, and reinforces trajectories that yield high rewards. This aligns with Generative Adversarial Imitation Learning (GAIL) (Ho & Ermon, 2016) but is adapted for the continuous flow matching setting.

A major challenge in applying policy gradients to flow models is that previous works typically view the denoising steps

**Algorithm 2** FAIL-PG: Flow Matching Adversarial Imitation Learning with Policy Gradient

1: **Input:** Expert images $x_E \sim \pi_E$, initial policy, reference policy and discriminator parameters $\theta_0, \theta_{\text{ref}}, \omega_0$
2: **for** iteration $i = 0, 1, 2, \ldots$ **do**
3:      Sample a batch of images $\{x_E^{(i)}\}$ from $x_E$
4:      For each prompt, sample $G$ images $\{x_0^{(i)}\} \sim \pi_{\theta_i}$
5:      Update discriminator from $\omega_i$ to $\omega_{i+1}$ with gradient:

$$\hat{\mathbb{E}}_{x_E^{(i)}}[\nabla_\omega \log \sigma(D_\omega(x_E^{(i)}))] + \hat{\mathbb{E}}_{x_0^{(i)}}[\nabla_\omega \log(1 - \sigma(D_\omega(x_0^{(i)})))]$$

6:      Compute rewards via updated discriminator $D_{\omega_{i+1}}$:

$$r(x_0^{(i)}) = -\log(1 - \sigma(D_{\omega_{i+1}}(x_0^{(i)})))$$

7:      Compute advantages within the $G$ samples group $\mathcal{G}_k$:

$$A(x_0^{(i)}) = \frac{r(x_0^{(i)}) - \mathbb{E}_{x_0' \in \mathcal{G}_k}[r(x_0')]}{\text{std}_{x_0' \in \mathcal{G}_k}(r(x_0'))}$$

8:      Set $\theta_{\text{old}} \leftarrow \theta_i$
9:      **for** epoch $e = 1, \ldots, E$ **do**
10:          Take a KL-constrained FPO step with gradient:

$$\hat{\mathbb{E}}_{x_0^{(i)}} \nabla_\theta \mathcal{L}_{\text{FPO}}(\theta) - \beta \nabla_\theta \text{KL}(\pi_\theta \| \pi_{\theta_{\text{ref}}})$$

11:      **end for**
12: **end for**

---

as a Markov Decision Process (MDP), requiring the transformation of the ODE into an SDE for exact likelihood computation. This is computationally expensive to estimate. To bypass this, we leverage **Flow Policy Optimization (FPO)** (McAllister et al., 2025), which utilizes the Conditional Flow Matching (CFM) loss to update the Evidence Lower Bound (ELBO). We approximate the likelihood ratio $r(\theta) = \frac{\pi_\theta(x)}{\pi_{\theta_{\text{old}}}(x)}$ via the difference in CFM losses and maximize the PPO-style (Schulman et al., 2017) clipped surrogate objective:

$$\begin{aligned} \mathcal{L}_{\text{FPO}}(\theta) &= \min\left(r(\theta)A(x), \text{clip}(r(\theta), 1 - \epsilon, 1 + \epsilon)A(x)\right), \\ r(\theta) &= \exp\left(\mathcal{L}_{\text{CFM}}(\theta_{\text{old}}, x) - \mathcal{L}_{\text{CFM}}(\theta, x)\right) \end{aligned} \quad (3)$$

To ensure stability during image generation, we enforce a constraint based on the KL divergence between the current policy and a reference policy (*e.g.*, the pre-trained model). Following the FPO derivation, this KL term is calculated as:

$$\text{KL}(\pi_\theta \| \pi_{\text{ref}}) = \mathbb{E}_{x \sim \pi_\theta} \left[ \mathcal{L}_{\text{CFM}}(\theta_{\text{ref}}, x) - \mathcal{L}_{\text{CFM}}(\theta, x) \right]. \quad (4)$$

As shown in Algorithm 2, FAIL-PG follows the core framework of GAIL but replaces the TRPO (Schulman et al., 2015) step with FPO. We adopt a PPO-style update rule and utilize Group Relative Policy Optimization (GRPO) (Shao et al., 2024) to normalize advantages.

Consequently, FAIL-PG can be viewed as an extension of GAIL to flow matching domain, providing a derivative-free policy optimization framework that maintains the generative quality of flow models while ensuring training stability.

## 2.3. Analysis of Optimization Strategies

**Gradient vs Score-Based Dynamics.** The core distinction between our methods lies in the density of the feedback signal. FAIL-PD computes pathwise derivatives $\nabla_x D(x)$, providing a dense, directional gradient that explicitly informs the policy how to deform the vector field. This preserves the structural smoothness of the flow manifold, ensuring stability and compatibility with inference-time constraints like Classifier-Free Guidance (CFG) (Ho & Salimans, 2022). Conversely, FAIL-PG relies on scalar rewards, acting as a "mode-seeking" operator that reinforces high-probability regions. While this enable rapid initial convergence by jumping to high reward modes, the lack of structural guidance results in higher variance and potential instability, i.e., model collapse, during extended training.

**Applicability.** These properties dictate distinct use cases. FAIL-PD is optimal choice for differentiable, white-box settings, offering the stability and precision required for large-scale training. FAIL-PG is strictly necessary for black-box or discrete scenarios where backpropagation is intractable, offering an efficient alternative when gradient access is restricted or fast convergence is a priority.

## 2.4. Stabilizing Training

Adversarial training is notoriously unstable, often suffering from mode collapse. These issues are exacerbated in high-dimensional image generation tasks. To stabilize training and accelerate convergence, we employ following strategies:

**Hybrid Imitation**. Following (Ren et al., 2024), we utilize a hybrid batch strategy—training on a mixture of online policy samples and expert demonstrations. During the policy update, we treat expert samples as "perfect" rollouts. This anchors the policy to the expert manifold, curtailing unnecessary exploration in the vast state space.

**Initialization and Warmup**. We apply a cold start by initializing the policy via Behavior Cloning on expert demonstrations. Furthermore, we implement a discriminator warmup phase, freezing the policy for the initial steps to ensure the discriminator provides a meaningful learning signal before the generator begins adversarial adaptation.

## 2.5. Discriminator Architecture Design

The choice of discriminator architecture is critical for stabilizing adversarial training. Instead of training from scratch, we leverage pre-trained representations to provide robust gradients. We investigate three distinct architectures:

**Visual Foundation Model (VFM)**. We employ pre-trained vision backbones (*e.g.*, CLIP (Lundberg & Lee, 2017), DINO (Oquab et al., 2023; Siméoni et al., 2025)) as feature extractors, adding a set of lightweight projection heads on top intermediate features. While efficient, this approach relies solely on visual features, potentially limiting its ability to enforce text-image alignment.

**Flow Matching (FM) Backbone**. To capture multi-modal dependencies, we utilize the pre-trained flow model itself as the discriminator backbone. We load the model's weights and attach a head to its intermediate feature maps and make all parameters trainable. This leverages the backbone's inherent understanding of the joint text-image distribution.

**Vision-Language Model (VLM)**. For advanced semantic discrimination, we adapt Large Multimodal Models by feeding both the image and text prompt as input. We extract the hidden states of the final token to feed the discriminator head. To bridge the gap between generation and discrimination tasks, we freeze the vision encoder but fine-tune the LLM parameters alongside the discriminator head.

## 3. Empirical Analysis

In this section, we conduct a comprehensive evaluation of FAIL, analyzing its properties from multiple perspectives. We aim to demonstrate its efficacy as a general post-training framework capable of converging to a target distribution.

### 3.1. Experimental Setup

**Expert Data Construction.** To ensure both high quality and semantic diversity in our target distribution, we curate prompts from a subset of HPDv3 (Ma et al., 2025) and UniGen-Bench (train-en-short split) (Wang et al., 2025a). We employ the frontier model, Gemini Pro 3 (Google, 2025), as our expert generator. Crucially, we generate only a single image per prompt. This constraint is designed to simulate realistic scenarios where obtaining multiple high-quality expert samples for a specific prompt is computationally expensive or infeasible. After filtering, we constructed a training dataset consisting of 13,000 prompt-image pairs.

**Implementation Details.** We adopt FLUX.1-dev (Labs, 2024) as our policy network. For the discriminator, we primarily utilize a VLM-based architecture initialized with Qwen3-VL-2B-Instruct (Bai et al., 2025a). To validate robustness, we also explore VFM and FM variants, instantiated with Dinov3b (Siméoni et al., 2025) and the original FLUX.1-dev backbone, respectively.

All models are optimized using AdamW (Loshchilov & Hutter, 2017) with a global batch size of 128. Training is distributed across 32 NVIDIA H20 GPUs, where each effec-

*Table 1.* Benchmark FAIL against preference Optimization methods using same expert data. URScore stands for UnifiedReward.

| Method | Steps | Unigen | DPG | HPSv3 | URScore |
|---|---|---|---|---|---|
| FLUX | - | 51.84 | 76.74 | 8.71 | 3.2965 |
| SFT | - | 58.22 | 80.34 | 9.28 | 3.3063 |
| RLHF-PD | 50 | 59.76 | 81.65 | 10.18 | 3.3406 |
| RLHF-PD | 400 | 57.97 | 80.35 | 8.90 | 3.2564 |
| RLHF-PG | 50 | 55.81 | 78.68 | 8.03 | 3.2287 |
| DPO-Online | 400 | 62.83 | 84.25 | 10.73 | 3.4183 |
| FAIL-PD (Ours) | 400 | 62.17 | 84.14 | 10.82 | 3.3938 |
| FAIL-PD (Ours) | 800 | 63.38 | 84.55 | 11.17 | 3.4468 |
| FAIL-PG (Ours) | 400 | 64.75 | 84.59 | 11.41 | 3.4438 |

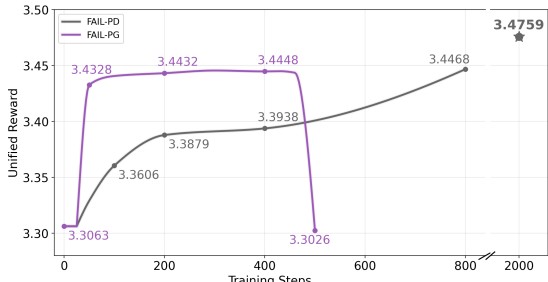

*Figure 2.* Convergence dynamics of FAIL variants. PG converges rapidly but suffers from collapse, PD shows long-term stability.

tive batch consists of 3 policy rollouts and 1 expert demonstration per prompt. Unless otherwise specified, ablations are conducted at $512 \times 512$ resolution for a single epoch (400 iterations), including a 25-step discriminator warmup phase. Comprehensive settings and detailed discriminator architectural specifications are provided in Appendix C.

**Evaluation Metrics.** We focus on two primary dimensions: **Prompt Following** and **Human Preference Alignment**. For prompt following, we utilize UniGen-Bench (Wang et al., 2025a), and DPG-Bench (Silver et al., 2014) to assess zero-shot generalization. For preference alignment, we employ the Alchemist (Startsev et al., 2025) dataset, reporting HPSv3 (Ma et al., 2025) for aesthetic quality and UnifiedReward (Wang et al., 2025b) for a holistic assessment of semantic and stylistic fidelity. Detailed evaluation protocols are provided in Appendix D. All samples are generated at $512 \times 512$ resolution with CFG disabled.

### 3.2. Comparison with Preference Optimization

We compare FAIL against standard preference optimization paradigms: Reinforcement Learning from Human Feedback (RLHF) and Direct Preference Optimization (DPO).

For **RLHF**, we train a specific reward model on expert-policy pairs and optimize the policy using both Policy Gradient (RLHF-PG) and Pathwise Derivative (RLHF-PD) estimators. For **DPO**, we implement an Online DPO baseline, which dynamically constructs preference pairs between expert demonstrations and current policy rollouts to ensure training stability. Specific implementation details for these baselines are detailed in Appendix D.

Intuitively, Online DPO can be viewed as a special case of FAIL-PG where the discriminator is assumed to be perfect (always distinguishing policy samples from expert samples) and provides a binarized reward signal.

**Analysis.** The quantitative results are presented in Table 1. We observe that standard RLHF exhibits instability characteristic of reward hacking. While the optimization successfully increased the reward values during training, the

ground-truth metrics degraded. Only early stage (50 steps) of RLHF-PD manages to outperform the SFT baseline. This highlights a critical bottleneck: with only 13,000 samples, the learned reward model is too weak to provide a robust direction toward the true high-quality distribution.

In contrast, FAIL achieves superior performance. Comparing Online DPO with FAIL-PG, we attribute FAIL's lead to two primary distinctions:

1. **Mitigating False Negatives:** Online DPO rigidly penalizes policy samples against expert data, creating "false negatives" when the policy generates high-quality images. FAIL utilize discriminator to dynamically judge the quality of samples, allowing the model to reinforce its own valid outputs rather than being arbitrarily rejected.

2. **Information Density:** While DPO compresses feedback into binary preference labels, FAIL utilizes continuous discriminator logits. By providing continuous feedback, FAIL ensures that every sample contributes richer information density than binary signal.

Finally, we notice FAIL-PD improve slower than FAIL-PG, it takes 800 steps to achieve similar performance. To understand if this performance gap persists over longer training horizons, we further investigate the long-term dynamics and stability of both methods in Section 3.3.

### 3.3. Convergence and Training Stability

To empirically validate the "Gradient vs. Score-Base" discussion in Section 2.3, we extend the training horizon to assess the long-term stability of both models. We use Unified Reward metric as it reflect the overall performance of model. The training dynamics are illustrated in Figure 2.

**Analysis.** The results shows a trade-off between convergence speed and stability of FAIL-PD and FAIL-PG:

1. **Mode-Seeking vs. Manifold Preservation:** Consistent with our analysis, FAIL-PG exhibits aggressive mode-seeking behavior, achieving rapid convergence with a high reward ($> 3.43$) within just 50 steps. However, this efficiency comes at the cost of stability: as training

*Table 2.* Impact of discriminator type. FAIL-PD benefit from stronger backbone while FAIL-PG is less sensitive to model choice.

| Method | Discriminator | Unigen | DPG | HPSv3 | URScore |
|---|---|---|---|---|---|
| FAIL-PD | Dinov3 | 60.65 | 82.78 | 10.28 | 3.4013 |
| FAIL-PD | FLUX.1 | 62.70 | 84.18 | 12.35 | 3.4474 |
| FAIL-PD | Qwen3vl | 62.17 | 84.14 | 10.82 | 3.3938 |
| FAIL-PG | Dinov3 | 64.23 | 84.50 | 11.30 | 3.4435 |
| FAIL-PG | FLUX.1 | 63.33 | 84.46 | 11.40 | 3.4496 |
| FAIL-PG | Qwen3vl | 64.75 | 84.59 | 11.41 | 3.4438 |

*Table 3.* Integrating FAIL with reward model acts as a regularizer, preventing the reward hacking observed in standalone RL.

| Method | Unigen | DPG | HPSv3 | URScore |
|---|---|---|---|---|
| FAIL-PD | 62.17 | 84.14 | 10.82 | 3.3938 |
| Reward Gradient | 58.50 | 82.15 | 12.21 | 3.4254 |
| Reward Gradient+FAIL-PD | 60.88 | 84.49 | 12.94 | 3.4832 |
| FAIL-PG | 64.75 | 84.59 | 11.41 | 3.4438 |
| FPO | 61.27 | 84.11 | 12.56 | 3.4503 |
| FPO+FAIL-PG | 64.27 | 84.70 | 11.75 | 3.4531 |

beyond 450 steps, the policy collapses, suggesting that scalar rewards alone may insufficient to constrain the policy to the flow manifold during extended optimization.

2. **Long-Term Stability:** In contrast, FAIL-PD demonstrates stability. While its initial improvement is more gradual, it maintains monotonic improvement well beyond the policy gradient method. By step 2,000, FAIL-PD achieves a higher performance (3.4759) without exhibiting signs of collapse. This confirms that the dense gradient signal preserves the vector field's structural integrity, allowing for deeper, more robust optimization.

These results suggest that while FAIL-PG is preferable for rapid, compute-efficient adaptation, FAIL-PD is essential for high-fidelity, long-horizon training where stability and peak performance are more important. To further improve the long-horizon stability of FAIL-PG, we introduce a simple KL masking strategy and provide its ablation in Appendix E. Additional visual analysis is provided in Appendix G.1.

### 3.4. Design Choice of Discriminator

We analyze the impact of the discriminator type on the policy performance. We select typical models for three distinct discriminator categories. The comparative results are presented in Table 2.

**Analysis.** For the FAIL-PD, the discriminator type significantly influences the final performance. The VFM baseline exhibits the lowest performance. We attribute this to two factors: 1) the lack of native text modality input, preventing the model from effectively judging text-image alignment, and 2) limited model capacity compared to other backbones, making the discriminator's gradients easier to hacking.

Conversely, the FM backbone achieves the best overall performance, particularly in preference alignment. We attribute this superior performance to two key advantages:

1. **Rich Feature Extraction:** FLUX is trained on massive high-quality image-text pairs. This extensive pre-training allows it to capture rich, dense semantic and aesthetic features, serving as a robust feature extractor that guides the policy toward a high-quality distribution.

2. **Latent Space Efficiency:** Unlike other discriminators

that operate in pixel space, the FM discriminator conducts judgments directly within latent space. This bypasses the computational overhead and potential information loss associated with VAE decoding, making the training process more efficient and stable.

In contrast to the PD variant, FAIL-PG is notably less sensitive to the choice of discriminator. Since FAIL-PG utilizes the discriminator as a black-box scalar reward function rather than backpropagating through its gradients, it mitigates the risk of the policy overfitting to specific artifacts in the discriminator's feature space. Consequently, performance is relatively consistent across different architectures.

### 3.5. Combine FAIL with reward model

A key distinction between FAIL and standard Reinforcement Learning (RL) approaches for flow matching is the source of the guidance signal: FAIL employs a *dynamically updated* discriminator to enforce distribution matching, whereas standard RL typically relies on a *static* reward model. This structural difference makes FAIL orthogonal to, and easily integratable with, existing RL frameworks.

To demonstrate this synergy, we combine FAIL with HPSv3-based reward optimization. We establish two baselines: **Reward Gradient** for the pathwise derivative setting and **Flow Policy Optimization** for the policy gradient setting. In these baselines, the discriminator is replaced by a frozen HPSv3 model. For the combined experiments:

- **FAIL-PD + Reward Gradient:** We directly sum the losses derived from the FAIL and the Reward Gradient.

- **FAIL-PG + FPO:** We combine the signals by adding the independently normalized scores from the FAIL discriminator and the HPSv3 before the policy update step.

**Analysis.** As shown in Table 3, the standalone Reward Gradient method exhibits characteristic reward hacking: while the HPSv3 score improves, metrics such as Unigen and DPG degrade significantly. Which aligns with observations from prior work such as REFL (Xu et al., 2023) that gradient optimization is easily hacked.

However, integrating FAIL-PD effectively mitigates this issue. The combined approach achieves the highest perfor-

*Table 4.* Applying FAIL-PG to discrete image generation substantially improve performance over the SFT baseline on Xomni.

| Method | Unigen | DPG | HPSv3 | URScore |
|---|---|---|---|---|
| Xomni | 28.32 | 73.08 | 6.84 | 3.1933 |
| Xomni SFT | 41.36 | 75.24 | 7.97 | 3.2501 |
| Xomni FAIL-PG | 54.24 | 80.94 | 9.91 | 3.3591 |

*Table 5.* FAIL generalizes effectively to video domain, yielding significant gains in semantic scores on VBench.

| Method | Overall | Quality Score | Semantic Score |
|---|---|---|---|
| Wan2.1 | 68.53 | 76.43 | 36.93 |
| Wan2.1 SFT | 73.02 | 79.71 | 46.27 |
| Wan2.1 FAIL-PD | 74.77 | 81.20 | 49.07 |
| Wan2.1 FAIL-PG | 75.71 | 81.70 | 51.75 |

mance in both HPSv3 (12.94) and UnifiedReward (3.4832). This indicates that FAIL functions as a potent regularizer. By introducing a dynamic distribution-matching objective alongside static reward maximization, FAIL prevents the policy from collapsing into narrow, high-reward modes (overfitting), thereby solving the reward hacking problem while simultaneously enhancing diversity. A similar performance also observed when combining FAIL-PG with FPO. See Appendix G.2 for visualization.

### 3.6. Generalize to other modality

To demonstrate universality, we extend FAIL to discrete image and video generation.

**Discrete Image Generation.** We utilize Xomni (Geng et al., 2025), applying FAIL-PG to the model's autoregressive LLM component to align token generation.

**Video Generation.** We apply FAIL to Wan2.1-1.3B (Wan et al., 2025), using prompts sampled from VidProM (Wang & Yang, 2024) and Wan2.2-A14B as expert. Performance is evaluated using the VBench (Huang et al., 2024).

Detailed experimental configurations for both modalities are provided in Appendix D.

**Analysis.** Table 4 demonstrates the effectiveness of FAIL in the discrete domain. Applying FAIL-PG to Xomni yields significant performance gains. Unlike continuous diffusion models, discrete generation models often suffer from the training-inference gap in next-token prediction frameworks. The use of an online training method like FAIL effectively alleviates this gap as we use online samples to update policy.

Furthermore, FAIL integrates seamlessly into video generation tasks without modification. As shown in Table 5, FAIL significantly improves the Semantic Score on VBench. We attribute the smaller relative gain in Quality Score to the absence of CFG during rollout. Video generation models typically rely heavily on large CFG scales for visual fidelity; optimizing samples in a CFG-free setting presents a greater

*Table 6.* Despite limited data, FAIL achieve a competitive results on DPG, demonstrating zero-shot generalization capabilities.

| Model | Global | Entity | Attribute | Relation | Other | Overall |
|---|---|---|---|---|---|---|
| Hunyuan-DiT (Li et al., 2024b) | 84.59 | 80.59 | 88.01 | 74.36 | 86.41 | 78.87 |
| Janus (Wu et al., 2025a) | 82.33 | 87.38 | 87.70 | 85.46 | 86.41 | 79.68 |
| Emu3-Gen (Wang et al., 2024) | 85.21 | 86.68 | 86.84 | 90.22 | 83.15 | 80.60 |
| FLUX.1 Dev (Labs, 2024) | 74.35 | 90.00 | 88.96 | 90.87 | 88.33 | 83.84 |
| SD3 Medium (Esser et al., 2024) | 87.90 | 91.01 | 88.83 | 80.70 | 88.68 | 84.08 |
| Janus-Pro-7B (Chen et al., 2025) | 86.90 | 88.90 | 89.40 | 89.32 | 89.48 | 84.19 |
| Z-Image-Turbo (Cai et al., 2025) | 91.29 | 89.59 | 90.14 | 92.16 | 88.68 | 84.86 |
| GPT-Image-1 High (OpneAI, 2025) | 88.89 | 88.94 | 89.84 | 92.63 | 90.96 | 85.15 |
| Seedream 3.0 (Gao et al., 2025) | 94.31 | 92.65 | 91.36 | 92.78 | 88.24 | 88.27 |
| Qwen-Image (Wu et al., 2025b) | 91.32 | 91.56 | 92.02 | 94.31 | 92.73 | 88.32 |
| FAIL-PD (Ours) | 85.00 | 91.65 | 90.22 | 92.52 | 91.62 | 87.32 |
| FAIL-PG (Ours) | 89.19 | 91.68 | 90.18 | 92.36 | 88.43 | 86.42 |

challenge, yet FAIL still achieve notable gain.

These results indicate that FAIL is a general-purpose preference optimization framework capable of generalizing effectively beyond standard diffusion-based image generation.

## 4. System Level Comparison

To validate the comprehensive capabilities of FAIL, we benchmark our framework against frontier text-to-image models, focusing on two dimensions of generation quality: prompt adherence and aesthetic fidelity. We compare FAIL against a wide range of competitive baselines. All FAIL-PD and FAIL-PG samples are generated using the same checkpoints across all benchmarks at a $512 \times 512$ resolution with optimal Classifier-Free Guidance (CFG).

**Prompt Adherence and Limitations.** We assess the model's ability to follow complex text instructions using DPG-Bench and UniGen-Bench. As shown in Table 6, FAIL significantly enhances the capabilities of its base model, FLUX.1 Dev. FAIL-PD achieves an overall DPG score of 87.32, surpassing the base model by 3.48 points and reaching parity with frontier models such as Seedream.

Table 7 presents evaluation results on Unigen. Trained on only 13,000 demonstrations, FAIL-PD achieves an overall score of 73.70, marking a substantial improvement over the FLUX.1 Dev baseline (61.61) and outperforming closed-source models like FLUX.1 Ultra and Imagen-3.0.

However, a closer inspection of the "Text" category in Table 7 reveals a critical insight regarding the boundaries of post-training alignment. While FAIL-PD improves text rendering performance from the baseline's 32.18 to 52.87, a significant gap remains compared to leading models like Qwen-Image (76.14). This suggests that while FAIL can effectively unlock and refine capabilities present in the latent space, it is ultimately bounded by the base model's pre-training. If the underlying character recognition or spelling capabilities are not robustly established during the pre-training stage, alignment methods alone cannot bridge the gap to frontier performance in these specific modalities.

**Aesthetic Fidelity.** In terms of human preference alignment, the results on the HPDv3 Benchmark (Table 8) demonstrate

*Table 7.* FAIL significantly outperforms the FLUX baseline and competitive frontier in Unigen Bench.

| Model | Action | Attribute | Compound | Layout | Grammar | Reasoning | Relationship | Style | Text | World Knowledge | Overall |
|---|---|---|---|---|---|---|---|---|---|---|---|
| Hunyuan-DiT (Li et al., 2024b) | 49.05 | 62.71 | 41.62 | 44.78 | 55.48 | 24.55 | 59.64 | 94.10 | 1.15 | 80.70 | 51.38 |
| FLUX.1 Dev (Labs, 2024) | 62.17 | 67.84 | 47.04 | 71.84 | 60.96 | 30.91 | 67.21 | 83.90 | 32.18 | 88.92 | 61.30 |
| Janus-Pro-7B (Chen et al., 2025) | 64.26 | 67.74 | 62.11 | 72.01 | 64.44 | 37.05 | 68.40 | 90.80 | 2.59 | 86.71 | 61.61 |
| SD3.5 Large (Esser et al., 2024) | 62.17 | 68.59 | 58.76 | 69.03 | 58.96 | 32.27 | 69.80 | 88.60 | 32.76 | 88.92 | 62.99 |
| FLUX.1 Ultra (Labs, 2024) | 76.50 | 76.50 | 67.78 | 81.54 | 70.70 | 43.18 | 77.54 | 90.60 | 37.36 | 91.61 | 70.67 |
| Z-Image-Turbo (Cai et al., 2025) | 69.30 | 74.54 | 63.03 | 78.36 | 64.57 | 39.68 | 71.57 | 90.00 | 70.69 | 92.25 | 71.40 |
| Imagen-3.0 (Baldridge et al., 2024) | 81.46 | 77.33 | 71.71 | 81.34 | 69.84 | 48.36 | 82.86 | 89.25 | 21.55 | 94.75 | 71.85 |
| FLUX-Kontext-Pro (Labs et al., 2025) | 77.66 | 79.20 | 72.68 | 84.47 | 72.69 | 55.68 | 79.34 | 94.78 | 50.29 | 91.61 | 75.84 |
| Qwen-Image (Wu et al., 2025b) | 84.13 | 87.61 | 73.32 | 85.52 | 60.29 | 53.64 | 79.70 | 95.10 | 76.14 | 94.30 | 78.81 |
| Seedream 3.0 (Gao et al., 2025) | 82.98 | 85.58 | 73.84 | 87.31 | 61.36 | 52.73 | 80.84 | 98.10 | 71.55 | 95.25 | 78.98 |
| FAIL-PD (Ours) | 74.81 | 79.59 | 70.62 | 83.02 | 65.11 | 44.04 | 80.20 | 94.50 | 52.87 | 92.25 | 73.70 |
| FAIL-PG (Ours) | 72.81 | 77.03 | 67.65 | 82.84 | 66.98 | 43.35 | 78.17 | 94.80 | 51.44 | 91.30 | 72.64 |

*Table 8.* FAIL achieve the highest overall score on HPDv3, confirming the effective internalization of the expert's aesthetics.

| Model | Characters | Arts | Design | Architecture | Animals | Natural Scenery | Transportation | Products | Plants | Food | Science | Others | All |
|---|---|---|---|---|---|---|---|---|---|---|---|---|---|
| Hunyuan-DiT (Li et al., 2024b) | 7.96 | 8.11 | 8.28 | 8.71 | 7.24 | 7.86 | 8.33 | 8.55 | 8.28 | 8.31 | 8.48 | 8.20 | 8.19 |
| Gemini 2.0 Flash | 9.98 | 8.44 | 7.64 | 10.11 | 9.42 | 9.01 | 9.74 | 9.64 | 9.55 | 10.16 | 7.61 | 9.23 | 9.21 |
| PixArt-Σ (Chen et al., 2024) | 10.08 | 9.07 | 8.41 | 9.83 | 8.86 | 8.87 | 9.44 | 9.57 | 9.52 | 9.73 | 10.35 | 8.58 | 9.37 |
| CogView4 (Zheng et al., 2024) | 10.72 | 9.86 | 9.33 | 9.88 | 9.16 | 9.45 | 9.69 | 9.86 | 9.45 | 9.49 | 10.16 | 8.97 | 9.61 |
| Infinity (Han et al., 2025) | 11.17 | 9.95 | 9.43 | 10.36 | 9.27 | 10.11 | 10.36 | 10.59 | 10.08 | 10.30 | 10.59 | 9.62 | 10.26 |
| Playground-v2.5 (Li et al., 2024a) | 11.07 | 9.84 | 9.64 | 10.45 | 10.38 | 9.94 | 10.51 | 10.62 | 10.15 | 10.62 | 10.84 | 9.39 | 10.27 |
| FLUX.1 Dev (Labs, 2024) | 11.70 | 10.32 | 9.39 | 10.93 | 10.38 | 10.01 | 10.84 | 11.24 | 10.21 | 10.38 | 11.24 | 9.16 | 10.43 |
| Kolors (Team, 2024) | 11.79 | 10.47 | 9.87 | 10.82 | 10.60 | 9.89 | 10.68 | 10.93 | 10.50 | 10.63 | 11.06 | 9.51 | 10.55 |
| FAIL-PD (Ours) | 11.91 | 10.16 | 10.30 | 12.22 | 11.29 | 10.96 | 12.18 | 11.92 | 11.70 | 12.00 | 9.59 | 11.40 | 11.28 |
| FAIL-PG (Ours) | 11.33 | 9.48 | 9.30 | 11.66 | 10.62 | 10.42 | 11.61 | 11.24 | 11.02 | 11.57 | 8.93 | 10.63 | 10.65 |

that FAIL-PD does not merely preserve the base model's quality but enhances it by a large margin. FAIL-PD achieves an overall score of 11.28, significantly outperforming the FLUX.1 Dev baseline of 10.43.

# 5. Related Work

## 5.1. Preference Optimization for Text-to-Image Models

While Supervised Fine-Tuning (SFT) on curated datasets (*e.g.*, Emu (Dai et al., 2023)) improves generation quality, it often fails to capture fine-grained human preferences. To address this, RLHF employing explicit Reward Models (RM) has been widely adopted. These approaches generally fall into two categories: Reward Gradient methods (Xu et al., 2023; Clark et al., 2023; Shen et al., 2025), which steer the model using reward score gradients, and Policy Gradient methods (Black et al., 2023; Xue et al., 2025b; Liu et al., 2025), which treat denoising as a sequential decision-making process to maximize expected reward.

To mitigate the instability and reward hacking associated with explicit reward modeling, Direct Preference Optimization (Rafailov et al., 2023) offers a stable alternative. Methods like Diffusion-DPO (Wallace et al., 2024) bypass the reward model, optimizing directly on preference pairs to implicitly maximize human alignment without the need for a explicit reward model.

## 5.2. Adversarial Training for Diffusion Models

Adversarial training has been explored in diffusion models mainly in two settings. One line of work augments diffusion training with GAN-style objectives to improve perceptual quality. Methods such as DiffusionGAN (Wang et al., 2022), DDGAN (Xiao et al., 2021), UFOGen (Xu et al., 2024), and SiDDM (Gong et al., 2023) introduce discriminators on generated samples or intermediate diffusion states, complementing diffusion objectives with gan loss.

Another line of work uses adversarial learning for diffusion model distillation. Approaches including ADD (Sauer et al., 2024b), LADD (Sauer et al., 2024a), and DMD2 (Yin et al., 2024) employ adversarial objectives to transfer knowledge from a multi-step diffusion teacher to a more efficient student, enabling high-quality generation with substantially fewer sampling steps.

## 5.3. Imitation Learning

Imitation Learning (IL) aim to train a policy that match the expert demonstrations. There are two board categories of IL approaches: behavioral cloning (BC) (Pomerleau, 1991b; Ross et al., 2011) and inverse reinforcement learning (IRL) (Ziebart et al., 2008; Abbeel & Ng, 2004). BC train a policy using regression to directly mimic the expert. While IRL jointly infer the policy and reward function.

Scaling IRL to large environments has been a major challenge, Adversarial Imitation Learning (AIL) offers a promising solution, AIL aim to directly match the state-action distributions of an agent and an expert through adversarial training. Generative adversarial imitation learning (GAIL) (Ho & Ermon, 2016) and its extensions (Fu et al., 2017b) train a generator policy to imitate expert and a discriminator to differentiate them, which resembles the idea of GANs.

Over the past years, many improvements (Orsini et al., 2021; Kostrikov et al., 2018) have been proposed to enhance GAIL's efficiency and robustness, and GAIL has been widely applied to various domains (Wulfmeier et al., 2024; Ye et al., 2025; Mao et al., 2025).

## 6. Limitation

While FAIL presents a robust framework for post-training alignment, there remains limitations. First, adversarial training introduces optimization complexities, despite our stabilization strategies, the training objective remains sensitive to hyperparameter and model design. Second, our experimental validation limited to a modest data size (13K), leaving the scaling of FAIL as an open question. Finally, the FAIL is ultimately bounded by the base model's pre-training, as evidenced by the persistent gap in text rendering performance, alignment methods like FAIL act as effective refiners but cannot synthesize fundamental capabilities that were not established during the pre-training stage.

## 7. Conclusion

We presented FAIL, a framework that formulates generative model post-training as adversarial imitation learning. FAIL overcomes the distribution shift problem of Supervised Fine-Tuning while eliminating the need for preference pairs or reward models required by RLHF and DPO. Our two optimization strategies—FAIL-PD for differentiable settings and FAIL-PG for discrete or black-box scenarios—achieve competitive performance using only 13,000 Gemini demonstrations. Beyond serving as a standalone alignment method, FAIL acts as an effective regularizer against reward hacking when combined with explicit reward optimization. The framework also generalizes to discrete image generation and video synthesis, establishing adversarial imitation learning as a versatile paradigm for generative alignment.

## Acknowledge

This work is supported by the Scientific Research Innovation Capability Support Project for Young Faculty (ZYGXQNJSKYCXNLZCXM-I22)

## Impact Statement

This paper introduces FAIL, a framework that democratizes high-quality generative alignment by eliminating the need for costly reward modeling and large-scale preference datasets. Beyond efficiency, our empirical results demonstrate that FAIL acts as a robust regularizer against "reward hacking", preventing models from exploiting metric vulnerabilities to generate high-scoring but non-sensical outputs. This contributes to the development of more reliable and semantically faithful generative systems.

However, because FAIL efficiently clones the target distribution, it is agnostic to the moral alignment of the expert data; any biases or harmful content in the reference demonstrations will be faithfully replicated by the student model. Additionally, the lowered computational barrier for generating hyper-realistic media raises concerns regarding potential misuse for deepfakes or misinformation. We strongly emphasize that practitioners must rigorously vet their expert data sources and employ safety filtering, as the safety of the aligned model is strictly bounded by the quality of the demonstrations it imitates.

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

## A. Derivation of the KL Divergence

In FAIL-PG, we employ a KL divergence constraint to prevent the policy $\pi_\theta$ from deviating exxcessively from the reference policy $\pi_{ref}$. The constraint is formulated as:

$$\text{KL}(\pi_\theta||\pi_{\text{ref}}) = \mathbb{E}_{x \sim \pi_\theta}\left[\mathcal{L}_{\text{CFM}}(\theta_{\text{ref}}, x) - \mathcal{L}_{\text{CFM}}(\theta, x)\right]. \tag{5}$$

The original FPO does not introduce this KL term. Here, we provide the brief derivation of this expression based on the conection between the Conditional Flow Matching(CFM) objective and the Evidence Lower Bound (ELBO).

### A.1. Relationship betwwen Log-Likelihood and CFM Loss

For Continuous Normalizing Flows (CNFs) trained via Flow Matching, the generative probability path is defined by a vector field $v_t$. As established by Kingma (Kingma & Gao, 2023) and utilized in recent reinforcement learning frameworks (McAllister et al., 2025; Xue et al., 2025a) for diffusion, the weighted flow matching loss $L_{CFM}(\theta, x)$ is mathematically equivalent to the negative ELBO of the data, up to a constant $C$ that only depends on the noise schedule and data distribution, but not on the learnable parameters $\theta$:

$$log\pi_\theta(x) \geq ELBO_\theta(x) = -L_{CFM}(\theta, x) + C. \tag{6}$$

In the context of Flow Policy Optimization (FPO), this ELBO is treated as a tight proxy for the true log-likelihood. Therefore, we approximate the log-probability of a sample $x$ under policy $\pi_\theta$ as:

$$log\pi_\theta(x) \approx -L_{CFM}(\theta, x) + C. \tag{7}$$

### A.2. Derivation of the KL Term

The Kullback-Leibler (KL) divergence between the current policy $\pi_\theta$ and the reference policy $\pi_{\text{ref}}$ is defined as:

$$D_{\text{KL}}(\pi_\theta||\pi_{\text{ref}}) = \mathbb{E}_{x \sim \pi_\theta}\left[\log \frac{\pi_\theta(x)}{\pi_{\text{ref}}(x)}\right] = \mathbb{E}_{x \sim \pi_\theta}\left[\log \pi_\theta(x) - \log \pi_{\text{ref}}(x)\right].$$

Substituting the ELBO approximation for the log-likelihoods of both the current policy and the reference policy:

$$\begin{aligned}
D_{\text{KL}}(\pi_\theta||\pi_{\text{ref}}) &\approx \mathbb{E}_{x \sim \pi_\theta}\left[(-\mathcal{L}_{\text{CFM}}(\theta, x) + C) - (-\mathcal{L}_{\text{CFM}}(\theta_{\text{ref}}, x) + C)\right] \\
&= \mathbb{E}_{x \sim \pi_\theta}\left[-\mathcal{L}_{\text{CFM}}(\theta, x) + C + \mathcal{L}_{\text{CFM}}(\theta_{\text{ref}}, x) - C\right] \\
&= \mathbb{E}_{x \sim \pi_\theta}\left[\mathcal{L}_{\text{CFM}}(\theta_{\text{ref}}, x) - \mathcal{L}_{\text{CFM}}(\theta, x)\right].
\end{aligned}$$

Crucially, because the constant $C$ is independent of the model parameters $\theta$, it cancels out in the difference. This allows us to estimate the KL divergence efficiently by comparing the flow matching losses of the generated sample $x$ under the reference weights $\theta_{\text{ref}}$ and the current weights $\theta$, without requiring expensive ODE integration to compute the exact likelihood change of variable.

## B. Discussion on One-step Denoising Approximation

In FAIL-PD, we avoid backpropagating through the full ODE trajectory and instead use the one-step denoising approximation in Eq.2. This section provides additional theoretical justification and empirical validation for this approximation.

### B.1. Derivation from the Flow Matching Interpolation Path

Recall that in flow matching, the linear interpolation path between a clean sample $x_0$ and Gaussian noise $\epsilon$ is defined as

$$x_t = (1 - t)x_0 + t\epsilon, \quad t \in [0, 1].$$

Given a current noisy state $x_t$, one Euler step using the learned vector field $v_\theta$ gives

$$x_{t+\Delta t} \approx x_t + \Delta t \cdot v_\theta(x_t, t).$$

Under the same linear interpolation parameterization, the state at time $t + \Delta t$ can also be written as

$$x_{t+\Delta t} \approx (1 - t - \Delta t)x_0' + (t + \Delta t)\epsilon,$$

*Table 9.* One-step error results.

| $[t_{\min}, t_{\max}]$ | PSNR ↑ | LPIPS ↓ |
|---|---|---|
| $[0, 0.5]$ | 35.17 | 0.030 |
| $[0.5, 1]$ | 31.13 | 0.115 |
| $[0, 1]$ | 32.55 | 0.083 |

where $x_0'$ denotes the clean sample predicted from this local one-step update. Solving for $x_0'$ yields

$$x_0' = \frac{x_t + \Delta t \cdot v_\theta(x_t, t) - (t + \Delta t)\epsilon}{1 - (t + \Delta t)}.$$

This is exactly the one-step denoising approximation used in FAIL-PD.

The approximation is local: rather than requiring the vector field to be globally linear over the entire trajectory, it only assumes that over a small step $\Delta t$, the Euler-updated state remains close to the same flow-matching interpolation path. This assumption is natural for rectified flow and flow matching models, where the training objective explicitly encourages straight interpolation between data and noise.

### B.2. Empirical Validation

To empirically evaluate the approximation quality, we measure the reconstruction error induced by the one-step approximation. We first generate 1,024 samples $x_0$ using the full 28-step solver. For each image, we sample $N = 4$ timesteps from a specified interval $[t_{\min}, t_{\max}]$, construct the corresponding noisy state $x_t = (1 - t)x_0 + t\epsilon$, recover $x_0'$ using Eq.2, and measure the error between $x_0'$ and $x_0$. This gives 4,096 total reconstruction pairs.

As shown in Tab.9, the one-step approximation achieves an overall PSNR of 32.55 and LPIPS of 0.083. The approximation is more accurate in the lower-noise region $[0, 0.5]$, but remains usable even in the higher-noise region $[0.5, 1]$. These results indicate that the local linearity assumption is reasonable for the flow matching trajectory used in our method.

## C. Detailed Experimental Settings

### C.1. Discriminator Architectures

Followed by (Kurach et al., 2019; Kumari et al., 2022), we design specific projection heads for different discriminator backbones to ensure effective gradient flow:

- **VFM and FM Variants:** For Dinov3b and FLUX.1-dev backbones, we uniformly extract intermediate features from the network layers. These features are processed by a discriminator head consisting of a LayerNorm, followed by a linear projection to logits with spectral normalization.

- **VLM Variant (Base):** For the Qwen3-VL-2B-Instruct backbone, we employ a streamlined architecture. We extract the hidden states of the EOS token from the final layer and map them directly to logits via a single linear layer.

### C.2. Global Configuration

Unless otherwise noted, all experiments utilize the global hyperparameters listed in Table 10. We utilize the AdamW optimizer for both the policy and the discriminator

### C.3. Task-Specific Learning Rates

We tune the learning rates for the policy ($lr_\theta$) and discriminator ($lr_\omega$) independently for each base model to ensure stability. Table 11 summarizes the specific learning rates used for the results reported in Section 3

## D. Detailed Evaluation and Baseline Settings

### D.1. Evaluation Protocols

Our evaluation framework is designed to capture diverse aspects of generation quality:

*Table 10.* **Global Hyperparameters.** These settings are shared across all tasks (Image, Video, Discrete) unless specified otherwise.

| Hyperparameter | Value |
|---|---|
| Optimizer | AdamW |
| Optimizer Momentum $(\beta_1, \beta_2)$ | $(0.9, 0.999)$ |
| Weight Decay | 0.0 |
| Gradient Accumulation Steps | 1 |
| Mixed Precision | BF16 |
| Gradient Clipping (Policy) | 1.0 |
| Gradient Clipping (Discriminator) | 1.0 |
| Global Batch Size | 128 |
| KL coefficient $\beta$ for FAIL-PG | 0.05 |

*Table 11.* Detailed Learning Rate Configuration. We independently tune the policy learning rate ($lr_\theta$) and discriminator learning rate ($lr_\omega$) for each algorithm and task to ensure stability.

| Method | Discriminator | Policy LR ($lr_\theta$) | Disc. LR ($lr_\omega$) |
|---|---|---|---|
| **Task: Image Generation** (Base: FLUX.1-Dev) | | | |
| FAIL-PD | Dinov3 | $1.0 \times 10^{-5}$ | $1.0 \times 10^{-5}$ |
| | FLUX.1 | $5.0 \times 10^{-6}$ | $1.0 \times 10^{-5}$ |
| | Qwen3-VL | $1.0 \times 10^{-5}$ | $2.0 \times 10^{-6}$ |
| FAIL-PG | Dinov3 | $1.0 \times 10^{-5}$ | $1.0 \times 10^{-5}$ |
| | FLUX.1 | $1.0 \times 10^{-5}$ | $2.0 \times 10^{-5}$ |
| | Qwen3-VL | $1.0 \times 10^{-5}$ | $2.0 \times 10^{-6}$ |
| **Task: Video Generation** (Base: Wan2.1-1.3B) | | | |
| FAIL-PD | Wan2.1 | $2.0 \times 10^{-6}$ | $1.0 \times 10^{-5}$ |
| FAIL-PG | Wan2.1 | $2.0 \times 10^{-6}$ | $1.0 \times 10^{-5}$ |
| **Task: Discrete Image Generation** (Base: Xomni-SFT) | | | |
| FAIL-PG | Dinov3 | $5.0 \times 10^{-6}$ | $5.0 \times 10^{-6}$ |

- **Prompt Following:** We utilize the *en-short* split of UniGen-Bench (Wang et al., 2025a), comprising 600 prompts synthesized by Gemini Pro 2.5. Evaluation is performed using the official offline pipeline powered by Qwen-72B (Bai et al., 2025b) for ablation, and official Gemini 2.5 Pro api for System Level Comparison. To assess zero-shot generalization, we additionally evaluate on DPG-Bench (Silver et al., 2014), which contains 1,065 diversity-focused prompts.

- **Human Preference Alignment:** We employ the Alchemist (Startsev et al., 2025) dataset, a collection of 3,350 prompts curated from web-scraped sources. We report HPSv3 (Ma et al., 2025) to measure aesthetic quality and UnifiedReward (Wang et al., 2025b) (specifically the *UnifiedReward-2.0-qwen3vl-8b* variant) to assess semantic alignment and coherence.

For inference, we generate 4 images per prompt for UniGen and DPG, and 1 image per prompt for Alchemist. All samples are generated with 28 steps at $512 \times 512$ resolution with Classifier-Free Guidance (CFG) disabled for ablation.

### D.2. Baseline Implementation Details

**RLHF Setup.** We first train a reward model utilizing the same architecture as our VLM discriminator. We construct preference pairs by treating expert data as the chosen response and policy rollouts as the rejected response, training the reward model for 1 epoch using the Bradley-Terry (BT) loss. Following reward modeling, we freeze the network and optimize the policy using the respective gradient estimators.

**Online DPO Setup.** We found standard DPO unstable when trained on fixed datasets in this domain. Therefore, we implement Online DPO, where we dynamically construct preference pairs during training. The current policy's rollout is treated as the rejected sample, and the corresponding expert data is treated as the chosen sample. This effectively simulates a "perfect" discriminator that always prefers the expert distribution.

**Discrete Image Generation Setup.** We utilize Xomni (Geng et al., 2025), a unified multimodal model capable of both

*Table 12.* Ablation of KL masking for FAIL-PG. KL masking improves long-horizon stability and allows FAIL-PG to train for 800 steps without collapse.

| Method | Steps | UniGen | DPG | HPSv3 | URScore |
|---|---|---|---|---|---|
| FAIL-PG | 400 | 64.75 | 84.59 | 11.41 | 3.4438 |
| FAIL-PG w/ KL masking | 800 | 65.49 | 84.81 | 11.45 | 3.4507 |

image understanding and generation. Xomni employs an LLM to autoregressively generate discrete image tokens, which are subsequently rendered into images via a diffusion decoder. We initialize the policy using Xomni weights prior to any RL alignment. Using the same dataset construction method as in our continuous image settings, we encode expert images into tokens and apply FAIL-PG to the LLM component. We employ DINOv3b (Siméoni et al., 2025) as the discriminator backbone, as our empirical results suggest FAIL-PG is robust to the choice of discriminator architecture in discrete settings.

**Video Generation Setup.** We evaluate video generation capabilities using the VidProM (Wang & Yang, 2024) dataset.

- **Dataset & Expert:** We sample a subset of 8,192 prompts from VidProM. We utilize Wan2.2-A14B (Wan et al., 2025) as the expert generator. For each prompt, we generate a single expert video at a resolution of $480 \times 480$ with a duration of 81 frames.

- **Policy & Training:** We use Wan2.1-1.3B as the student policy. The discriminator architecture mirrors the Flow Matching backbone setting used in our image experiments, adapted to process the spatiotemporal latents of the video model.

## E. KL Masking for Stabilizing FAIL-PG

As discussed in Sec.3.3, FAIL-PG converges rapidly but may become unstable under extended training. Although the KL regularization term in Eq.4 constrains the policy toward the reference model on average, we find that a small number of high-KL samples can still dominate the policy update. This issue is particularly pronounced in FAIL-PG because the discriminator provides scalar rewards rather than directional gradients.

To improve long-horizon stability, we introduce a simple KL masking strategy. For each generated sample $x_i$ in a batch, we compute its per-sample KL proxy, then define a batch-adaptive threshold:

$$T = \text{mean}(\text{KL}_i) + c \cdot \text{std}(\text{KL}_i),$$

where $c$ is a hyperparameter. Unless otherwise specified, we set $c = 2$. During the policy update, samples with $\text{KL}_i > T$ are excluded from the FAIL-PG objective. The resulting masked objective is:

$$L_{\text{mask}}(\theta) = \frac{\sum_i \mathbb{1}[\text{KL}_i \leq T] \, L_{\text{FPO}}(\theta; x_i)}{\sum_i \mathbb{1}[\text{KL}_i \leq T]}$$

This masking procedure removes samples that drift unusually far from the reference policy, while preserving the standard FAIL-PG update for the majority of the batch. Unlike tuning a fixed KL coefficient, the threshold adapts to the KL distribution of each batch and does not require reward-scale calibration.

Tab.12 shows that KL masking stabilizes FAIL-PG under a longer training horizon. Compared with the 400-step FAIL-PG baseline, training with KL masking for 800 steps further improves UniGen, DPG, HPSv3, and UnifiedReward. These results suggest that KL masking is an effective and lightweight stabilization technique for extending FAIL-PG training.

## F. Additional Ablation Studies

We provide additional ablations to analyze the effect of expert data scale, training cost, cold-start initialization, and distributional alignment.

### F.1. Data Scaling.

Since FAIL is an on-policy method, most training samples are generated by the current policy. Therefore, the training horizon and expert-data quality may affect performance as much as the raw expert-data size. Due to compute constraints, we evaluate smaller expert sets of 4K and 8K samples under the same training budget.

*Table 13.* Ablation on expert data scale. Increasing expert data mainly improves prompt-following metrics, while preference metrics change more mildly.

| Method | Expert Data | Steps | UniGen | DPG | HPSv3 | URScore |
|---|---|---|---|---|---|---|
| FAIL-PD | 4K | 800 | 60.04 | 83.32 | 11.04 | 3.4379 |
| FAIL-PD | 8K | 800 | 62.29 | 84.32 | 11.02 | 3.4458 |
| FAIL-PD | 13K | 800 | 63.38 | 84.55 | 11.17 | 3.4468 |
| FAIL-PG | 4K | 400 | 63.36 | 84.42 | 11.34 | 3.4433 |
| FAIL-PG | 8K | 400 | 63.63 | 84.56 | 11.31 | 3.4436 |
| FAIL-PG | 13K | 400 | 64.75 | 84.59 | 11.41 | 3.4438 |

*Table 14.* Wall-clock comparison per training step.

| Phase | FAIL-PD (s) | FAIL-PG (s) |
|---|---|---|
| Rollout | 39.88 | 40.73 |
| Discriminator update | 0.79 | 1.04 |
| Policy update | 12.06 | 50.08 |
| Total | 52.73 | 91.85 |

As shown in Tab.13, increasing expert data improves UniGen and DPG, while HPSv3 and URScore are relatively stable. FAIL-PD benefits more clearly from additional data, which is consistent with its stronger long-horizon scalability observed in Sec.3.3.

## F.2. Wall-clock Comparison.

We report the per-step wall-clock cost in Tab.14. All methods are measured under the same hardware and training configuration.

The discriminator update only contributes a small fraction of the total cost. The main overhead comes from rollout and policy optimization, which is shared by on-policy optimization methods.

## F.3. Impact of Cold Start.

When the expert distribution differs substantially from the initial policy, directly applying adversarial or preference-based optimization can be unstable. To study this effect, we compare four settings: the original FLUX model, the SFT cold-start model, FAIL-PD trained directly from FLUX, and FAIL-PD initialized from the SFT checkpoint. Both FAIL-PD variants are trained for 800 steps under the same adversarial training setup.

As shown in Tab.15, SFT already improves over the original FLUX model, especially on prompt-following metrics such as UniGen and DPG. This indicates that behavior cloning provides a useful initialization when the expert distribution is far from the base policy. Without this cold start, FAIL-PD can still improve preference-oriented metrics such as HPSv3 and URScore, suggesting that the adversarial signal can guide the model toward the target distribution. However, its UniGen score drops from 63.38 to 55.23, showing that direct adversarial training under a large distribution shift can hurt prompt-following alignment. In contrast, initializing FAIL-PD from the SFT checkpoint achieves the best overall trade-off, improving both prompt-following and reward-based metrics.

In our experiments, directly applying FAIL-PG or Online DPO without a cold start is less stable and can collapse within roughly 100 steps, further highlighting the importance of initialization for policy-gradient-style training.

## F.4. Distribution Distance Analysis.

To further quantify whether FAIL reduces the distribution gap between policy outputs and expert demonstrations, we measure distribution distances in a semantic feature space. Specifically, we use 1,024 expert samples and generate policy rollouts using the same prompts. We extract image features with SigLIPv2 so400m (Tschannen et al., 2025) and compute the Maximum Mean Discrepancy (MMD) (Gretton et al., 2012) and Sliced Wasserstein Distance (SWD) (Karras et al.,

*Table 15.* Impact of cold-start initialization and adversarial fine-tuning.

| Method | Steps | UniGen | DPG | HPSv3 | URScore |
|---|---|---|---|---|---|
| FLUX | – | 51.84 | 76.74 | 8.71 | 3.2965 |
| SFT | – | 58.22 | 80.34 | 9.28 | 3.3063 |
| FAIL-PD without cold start | 800 | 55.23 | 81.00 | 11.28 | 3.4378 |
| FAIL-PD with cold start | 800 | 63.38 | 84.55 | 11.17 | 3.4468 |

*Table 16.* Distribution distance between policy samples and expert samples. Lower is better.

| Method | Steps | MMD $\times 10^3 \downarrow$ | SWD $\times 10^3 \downarrow$ |
|---|---|---|---|
| FLUX | - | 18.26 | 7.10 |
| SFT | - | 9.60 | 5.19 |
| FAIL-PG | 400 | 2.52 | 2.81 |
| FAIL-PD w/o cold start | 800 | 7.54 | 4.72 |
| FAIL-PD | 800 | 3.84 | 3.42 |
| FAIL-PD | 2000 | 2.24 | 2.76 |

2018) between the two feature sets. For MMD, we use RBF kernels with bandwidths $\sigma \in \{1, 2, 4, 8\}$. Lower values indicate that the generated distribution is closer to the expert distribution.

Tab.16 shows that SFT substantially reduces the distribution distance from the expert data compared with the original FLUX model, decreasing MMD from 18.26 to 9.60 and SWD from 7.10 to 5.19. FAIL further reduces this gap. FAIL-PG achieves strong distribution alignment at 400 steps, which is consistent with its fast convergence behavior.

For FAIL-PD, cold-start initialization is again important. Without a cold start, FAIL-PD improves over FLUX and SFT in terms of distribution distance, but remains worse than the fully initialized version. With cold start, FAIL-PD reduces MMD to 3.84 and SWD to 3.42 after 800 steps. Extending training to 2,000 steps further reduces the distances to 2.24 and 2.76, respectively. These results support our interpretation that FAIL-PD provides a stable long-horizon optimization signal: it continues to move the policy distribution toward the expert distribution while maintaining training stability.

# G. Visualization Analysis

### G.1. Visualization of Training Dynamics

We visualize the training dynamics discussed in Section 3.3 to qualitatively assess convergence behavior. All samples are generated without Classifier-Free Guidance (CFG). For FAIL-PD, we display generation results at steps 100, 800, and 2,000 in Figure 4. Consistent with our analysis, FAIL-PD demonstrates monotonic improvement as training progresses, with the student distribution progressively aligning with the expert demonstrations in terms of both style and structure.

Given the rapid convergence of FAIL-PG, we present results at steps 0 (SFT baseline), 100, and 400 in Figure **??**. We observe distinct phases of improvement: between steps 0 and 100, the model achieves a rapid boost in overall visual quality and semantic alignment. Subsequently, from step 100 to 400, the optimization shifts focus to refining low-level details, resulting in significant improvements in text rendering and fine-grained textures.

### G.2. Visualization of Reward Model Integration

To further validate FAIL's capability as a regularizer against reward hacking, we compare the qualitative results of standalone FAIL-PD, standalone Reward Gradient (optimizing HPSv3 (Ma et al., 2025)), and their combination in Figure 5.

As observed in the first column, optimizing HPSv3 alone leads to severe reward hacking, characterized by oversaturation and unnatural high-frequency artifacts. However, combining this objective with FAIL (second column) effectively mitigates these phenomena. The combined approach preserves the structural integrity and naturalness of the images while successfully inheriting the aesthetic improvements from the HPSv3 reward signal.

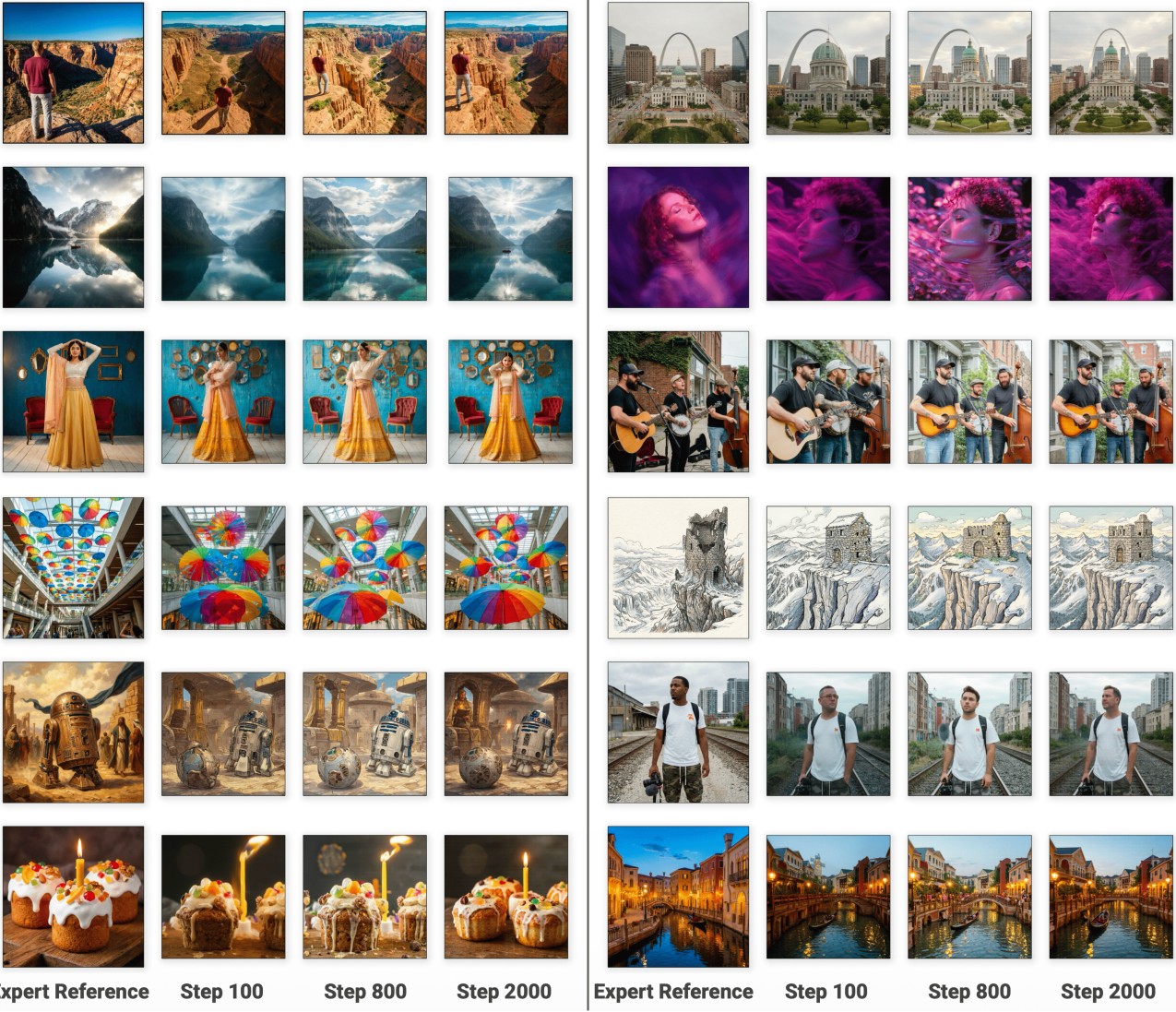

*Figure 3.* The visualization results of FAIL-PD in different training steps. FAIL-PD show consistence improvement and distribution alignment to the expert demonstrations.

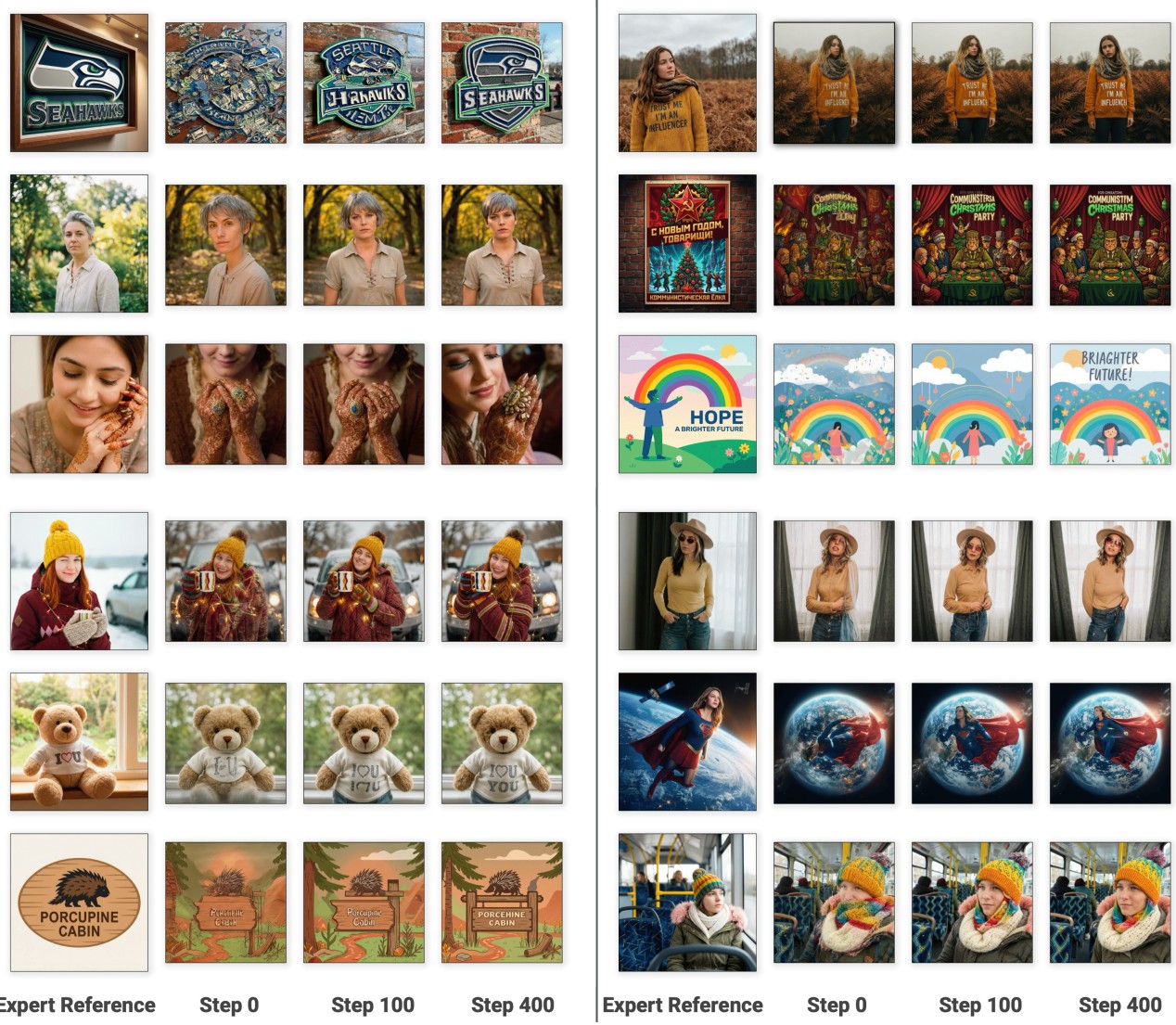

*Figure 4.* FAIL-PG convergence rapidly, it improve the overall quality first, then optimize the fine-grained detail.

| Reward Gradient | Reward Gradient + FAIL | FAIL | Reward Gradient | Reward Gradient + FAIL | FAIL |
| --- | --- | --- | --- | --- | --- |

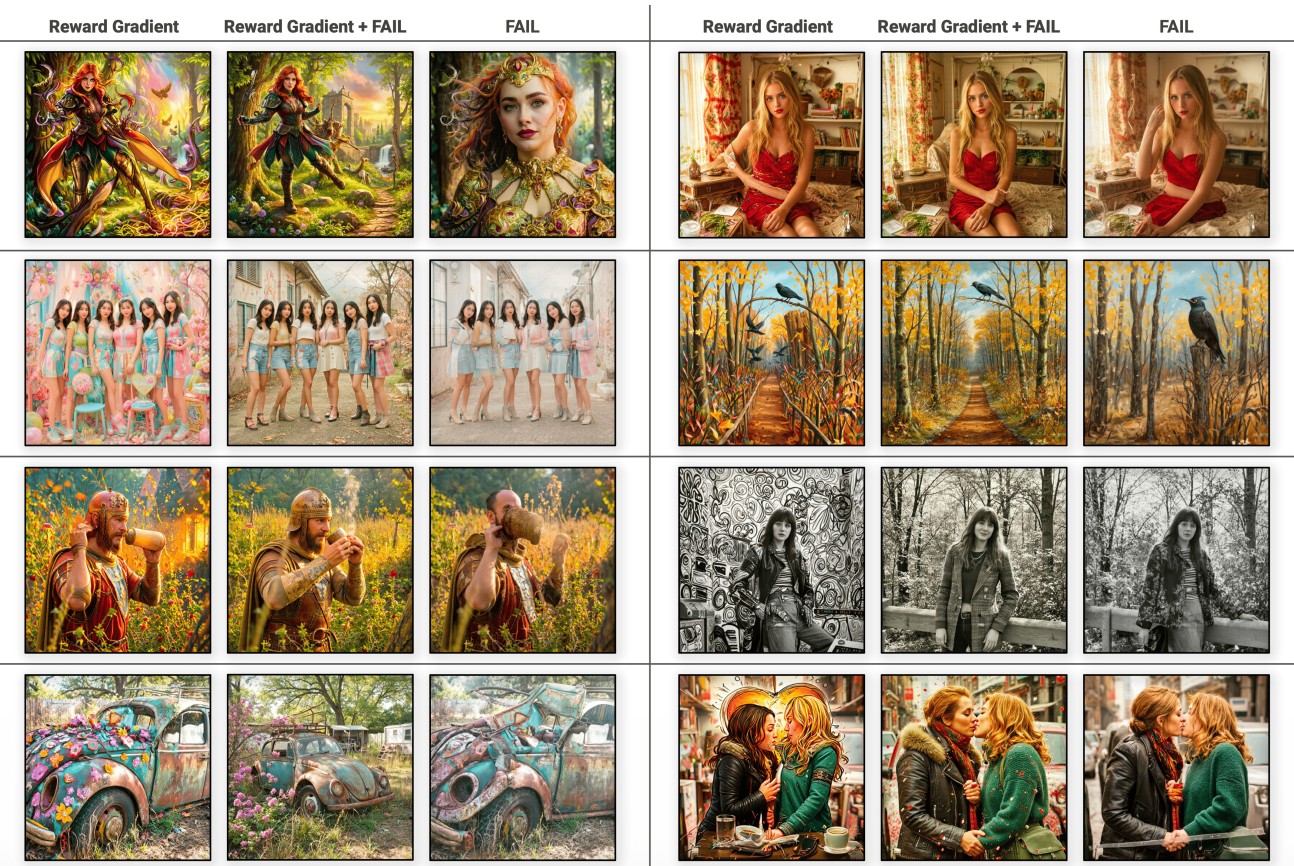

*Figure 5.* Integrate FAIL with reward model could alleviate the reward hacking phenomenon.

