# OpenReview forum: "FAIL: Flow Matching Adversarial Imitation Learning for Image Generation"
_ICML.cc/2026/Conference — ICML 2026 regular_

### Official Review · Reviewer_8XyN · 2026-03-10

**Soundness:** 2
**Presentation:** 3
**Significance:** 3
**Originality:** 3
**Overall Recommendation:** 4
**Confidence:** 3

**Summary:**

This work introduces Flow Matching Adversarial Imitation Learning (FAIL), a framework for aligning image generative models with target distributions using expert demonstrations rather than pairwise preferences. To reduce the drift of a learnable policy, the feedback signal from a discriminator is utilized. Two policy optimization strategies are introduced: FAIL-PD, a white-box approach leveraging differentiable ODE solvers, and FAIL-PG, a black-box alternative for discrete or gradient-intractable scenarios. Experiments are conducted on DPG-Bench and UniGen-Bench, and the proposed method shows improved prompt following and aesthetic quality while using a dataset of 13,000 expert images.

**Compliance With Llm Reviewing Policy:**

Affirmed.

**Final Justification:**

This work introduces an interesting model alignment framework for target distribution generation. In the rebuttal, the authors clarified certain misunderstandings and performed additional experiments to address the raised concerns such as stability of the method for large distributional shifts, scalability and the dataset used within this study. Based on the rebuttal, I remain with my initial decision of a weak accept and encourage the authors to add the additional experiments and comments to the revised paper for clarity and completeness.

**Key Questions For Authors:**

1. How does the model behave with larger demonstration sets? Given the sensitivity of adversarial training, does the performance gain plateau quickly, or does the framework benefit linearly from more "expert" data?
2. If the expert demonstrations are significantly different from the base model's pre-training distribution (new domain such as agricultural images), how does the proposed method handle the distribution shift? Does the model tend to "hallucinate" to match the expert, or does it retain the base model's structural priors?
3. It is mentioned that FAIL-PG suffers from mode-seeking behavior over long horizons. Are there specific regularization techniques that could be integrated to stabilize the policy gradient variant?

**Limitations:**

Yes

**Strengths And Weaknesses:**

## Strengths
1. By eliminating the need for expensive pairwise preference data and explicit reward models, the proposed method removes a major bottleneck in the alignment pipeline.
2. The usage of a continuous discriminator signal is innovative. Such modifications already exist for diffusion distillation and improving the convergence of diffusion models but it's application to model alignment is interesting.
3. Experimental validation of the proposed approach across applications  and diverse modalities, including continuous diffusion, discrete autoregressive image generation (Xomni), and video generation (Wan2.1) is commendable.
4. Empirical performance against SoTA models on UniGen-Bench highlights the framework's practical utility.

## Weakness
1. As is common with adversarial frameworks, the proposed method is highly sensitive to hyper-parameter selection and discriminator architecture. The paper provides stabilization strategies, but the question of whether this framework can scale to much larger datasets without losing stability remains unanswered.
2. The proposed framework can be understood as a refiner rather than a corrector. As noted in the paper, if the base model lacks certain characteristics (e.g., specific character recognition or spelling), the method cannot synthesize these missing features during post-training alignment.
3. Given the smaller dataset used for model alignment and the significant improvement in scores across benchmarks, a question regarding dataset bias is raised. The prompts are constructed from the benchmarks to generate the expert images and the results are again compared using the same benchmark.

---

> ### Author Rebuttal · Authors · 2026-03-30
>
> We thank the reviewer for the thoughtful review and helpful feedback. Below we address the main questions.
>
> > Q1: The scalability of FAIL.
>
> Due to resource limits, we did not run experiments beyond 13K expert samples. However, Fig.2 shows that FAIL-PD continues to improve as training proceeds, which suggests better scaling potential, whereas FAIL-PG converges rapidly and is more suitable for compute-constrained settings.
>
> **We also conducted an ablation over training data size (4K / 8K / 13K). Please see our response to Reviewer uo5X, Q1.**
>
> > Q2: How to handle large distribution shifts.
>
> When the expert distribution differs substantially from the current policy, a **cold start** is important for both stability and final performance. In our experiments, directly applying FAIL-PG or Online DPO without a cold start leads to collapse within roughly 100 steps. FAIL-PD can still move toward the target distribution without a cold start, but prompt-following performance degrades substantially. The results are shown in Tab.5.
>
> **Tab.5 Ablation of cold start**
> | Method | Steps | Unigen | DPG | HPSv3 | URScore |
> | :--- | :--- | :--- | :--- | :--- | :--- |
> | FLUX | - | 51.84 | 76.74 | 8.71 | 3.2965 |
> | SFT | -| 58.22 | 80.34 | 9.28 | 3.3063 |
> | FAIL-PD without cold start | 800 | 55.23 | 81.00 | 11.28 | 3.4378 |
> | FAIL-PD with cold start | 800 | 63.38 | 84.55 | 11.17 | 3.4468 |
>
> To further analyze the distribution shift, we measured the **Maximum Mean Discrepancy (MMD)** and **Sliced Wasserstein Distance (SWD)** between policy outputs and expert data. Using 1,024 expert samples and corresponding policy rollouts for the same prompts, we first extracted features with SigLIPv2-so400m and then computed MMD and SWD between the two feature sets. For MMD, we used an RBF kernel with bandwidths $\sigma \in \{ 1, 2, 4, 8 \} $.
>
> **Tab.6 Distribution distance analysis of FAIL**
> | Method | Steps | MMD $\times 10^3$ | SWD $\times 10^3$ |
> | :--- | :--- | :--- | :--- |
> | FLUX | - | 18.26 | 7.10 |
> | SFT | - | 9.60 | 5.19 |
> | FAIL-PG | 400 | 2.52 | 2.81 |
> | FAIL-PD without cold start | 800 | 7.54 | 4.72 |
> | FAIL-PD | 800 | 3.84 | 3.42 |
> | FAIL-PD | 2000 | 2.24 | 2.76 |
>
> As shown in Tab.6, SFT largely reduces the distance between the policy and expert distributions, and FAIL reduces it further. Without a cold start, FAIL-PD still improves over the SFT baseline, but remains clearly worse than the fully initialized version. This highlights the importance of a cold start under large distribution shift.
>
> > Q3: Concern about dataset bias.
>
> Our training prompts are drawn from subsets of **HPDv3** and the **UniGen-Bench train-en-short split**, and there is **no overlap** with the evaluation prompts. In our ablations, UniGen can be viewed as the same benchmark family but with non-overlapping evaluation prompts, while **DPG** is out-of-domain. For human-preference evaluation, we use **Alchemist**, which is also out-of-domain and has a different prompt distribution from our training data.
>
> > Q4: Strategy to stabilize FAIL-PG.
>
> We conducted additional analysis of the FAIL-PG collapse phenomenon and tested simple KL masking stabilization strategies. In particular, we successfully extended FAIL-PG to 800 steps. **Please see our responses to Reviewer uo5X, Q2 and Reviewer BqcN, Q4.**

---

> > ### Author Rebuttal · Reviewer_8XyN · 2026-04-02
> >
> > The rebuttal by the authors have resolved many of the raised concerns. I would tends towards the initial decision of a weak accept.

---

> > > ### Author Response · Authors · 2026-04-04
> > >
> > > Thank you for the update and for carefully considering our rebuttal. We are glad that the additional clarifications helped address your concerns. We sincerely appreciate your thoughtful review, your encouraging assessment, and your time.

---

### Official Review · Reviewer_BqcN · 2026-03-13

**Soundness:** 3
**Presentation:** 3
**Significance:** 3
**Originality:** 2
**Overall Recommendation:** 4
**Confidence:** 2

**Summary:**

The paper focuses on post-training alignment of flow matching models using Generative Adversarial Imitation Learning (GAIL). By casting the fine-tuning problem as GAIL, the authors derive two practical algorithms: FAIL-PD, which updates the generative policy using pathwise derivatives through the dynamically updated discriminative reward, and FAIL-PG, which uses Flow Policy Optimisation (FPO) to implement PPO-style policy gradient updates in settings where FAIL-PD is computationally infeasible or the sampling procedure is non-differentiable. The authors conduct a wide range of experiments demonstrating that FAIL achieves stronger performance than SFT, RLHF, and DPO, performs competitively on the Text-to-Image benchmark, and could be combined with static rewards to mitigate reward hacking.

**Compliance With Llm Reviewing Policy:**

Affirmed.

**Final Justification:**

I maintain my assessment of 'Weak Accept': the paper addresses a practically important problem with thorough experimental validation, and the rebuttal resolved my technical concerns. While I appreciate the argument that the novelty lies at the framework level, in my opinion, the core algorithmic contributions at the method level are not entirely novel over existing adversarial training and FPO methods.

**Key Questions For Authors:**

1. Could you provide a more thorough analysis of the one-step approximation used for training the generative policy using FAIL-PD?
2. Why does FAIL-PG exhibit mode-seeking behaviour? Why doesn't the introduction of the KL regularisation term help, and the training still collapses?

**Limitations:**

Yes

**Strengths And Weaknesses:**

### Strengths

1. The experimental validation is thorough and includes continuous, discrete image generation and video generation setups, showing the generalisability of the proposed FAIL-PD and FAIL-PG methods in the limited expert data scenario across different domains.
2. The dynamic reward modelled via the discriminator is conceptually appealing. Due to its nature, it can be used in conjunction with static rewards as regularisation against reward hacking, which is practically relevant.
3. Training adversarial methods is genuinely difficult due to their unstable nature, so I believe there is a significant practical merit in the authors' ablation of the discriminator's design for the FAIL-PD setup.

### Weaknesses
1. The framing of post-training as imitation learning is interesting and aids presentation. But at its core, FAIL-PD is a GAN objective applied to single-step approximated images with gradients through the denoising step, which is not novel.
2. The proposed FAIL-PD introduces a one-step approximation, which is not rigorously justified, and the paper would benefit from a more thorough analysis of how the approximation error depends on $t$ and whether a few-step approach would yield better results.

---

> ### Author Rebuttal · Authors · 2026-03-30
>
> We thank the reviewer for the thoughtful review and helpful feedback. Below we address the main questions.
>
> > Q1: Theoretical analysis of the one-step denoising approximation.
>
> Our one-step approximation is derived directly from the Flow Matching linear interpolation path.
>
> Given
>
> $x_t = (1-t)x_0 + t \epsilon$
>
> and one Euler step
>
> $x_{t+\Delta t} = x_t + \Delta t \cdot v_{\theta}(x_t, t)$
>
> we write
>
> $x_{t+\Delta t} \approx (1-t-\Delta t) x_0' + (t + \Delta t ) \epsilon$
>
> And solve for the predicted clean sample $x_0'$ which yields Eq.(2). The approximation is local: over a small $\Delta t$, we assume that $x_{t+ \Delta t}$ is still well approximated by the same interpolation. This is reasonable because $\Delta t$ is small, and Flow Matching uses straight interpolation paths by construction. Related one-step recovery schemes have also been used in recent reward-gradient methods such as SRPO [1].
>
> > Q2: Empirical analysis of the approximation error.
>
> To validate the local-linearity assumption, we measured reconstruction error as follows. We first generated 1,024 images $x_0$ using the full 28-step solver. For each image, we sampled $N=4$ timestep in a given interval [$t_{min}$, $t_{max}$], constructed $x_t$, recovered $x_0'$ using Eq.(2), and measured the error between $x_0'$ and $x_0$. This gives 4,096 total samples. We report PSNR and LPIPS in Tab.4.
>
> **Tab.4 Error Analysis for 1-step Approximation**
> | [$t_{min}$, $t_{max}$] | PSNR $\uparrow$ | LPIPS $\downarrow$ |
> | :--- | :--- | :--- |
> | [0, 0.5] | 35.17 | 0.030 |
> | [0.5, 1] | 31.13 | 0.115 |
> | [0, 1] | 32.55 | 0.083 |
>
> For reference, the Qwen-Image report [2] lists VAE reconstruction PSNRs of 32.84 (FLUX), 31.22 (SD-3.5), and 33.42 (Qwen-Image). Our overall PSNR of 32.55 is therefore within a comparable range. Even for large-noise samples with $t \in [0.5, 1]$, the approximation error remains moderate.
>
> Regarding the few-step alternative, as we mentioned in the response in **Reviewer RJj3 Q1**, previous works have conduct the ablation and conclude using 1 step backpropagation achieves highly competitive results.
>
> > Q3: Why does FAIL-PG exhibit mode-seeking behavior?
>
> FAIL-PG is more mode-seeking because it uses the discriminator only as a **scalar reward**, not as a source of **directional gradients**. The update therefore reinforces samples that happen to receive high reward, which naturally concentrates probability mass on a smaller set of high-reward modes.
>
> > Q4: Why does KL regularization not fully prevent collapse?
>
> In fact, KL regularization does help: in our experiments it extends stable training from roughly 150 steps to about 450 steps. The later collapse likely comes from two factors. First, policy-gradient updates have high variance; in our preliminary experiments, using FPO with a static reward model also eventually collapsed with long training. Second, using KL only as an auxiliary loss may be insufficient: if an OOD low-quality sample receives a spuriously high discriminator reward, the policy-loss term can still amplify it enough to destabilize training, even if the KL term penalizes it.
>
> Two practical mitigations are:
>
> 1. increasing the rollout count to reduce variance; and
>
> 2. incorporating KL directly into the reward,
>
> $r_{adj}(x) = r_{disc}(x) - \beta \cdot D_{KL}(x)$
>
> as in KL-regularized PPO-style optimization.
>
> We also tested a simple KL masking strategy that removes samples with unusually large per-sample KL from the policy update. **This keeps FAIL-PG stable through 800 steps; please see our response to Reviewer uo5X, Q2.**
>
> [1] SRPO: Directly Aligning the Full Diffusion Trajectory with Fine-Grained Human Preference
>
> [2] Qwen-Image Technical Report

---

> > ### Author Rebuttal · Reviewer_BqcN · 2026-04-03
> >
> > The rebuttal adequately addressed the raised concerns. I find the empirical analysis of the one-step denoising approximation error sufficiently convincing, and the discussion regarding KL regularisation in FAIL-PG comprehensive. The proposed method solves an important post-training problem and has significant practical relevance. However, in my opinion, the methods themselves (FAIL-PD, FAIL-PG) are not entirely novel, with FAIL-PD effectively using adversarial training for fine-tuning and FAIL-PG extending the previous FPO method with dynamic discriminator rewards. Therefore, I am inclined to maintain my initial score of weak accept.

---

> > > ### Author Response · Authors · 2026-04-04
> > >
> > > Thank you again for the thoughtful follow-up and acknowledging that our rebuttal addressed the technical concerns. We also sincerely appreciate your positive assessment that the paper tackles an important post-training problem and has clear practical relevance.
> > >
> > > On originality, we agree that some components of FAIL-PD/FAIL-PG are related to existing adversarial training and policy optimization ideas when viewed in isolation. Our intended novelty is primarily at the **framework level**: we formulate flow-matching post-training as an **imitation-learning problem** and show that this perspective leads to a simple and practical alignment approach for FM models. The simplicity is intentional—we wanted to validate the idea in a form that can be easily integrated into existing FM post-training pipelines, while also helping mitigate reward hacking through the dynamic discriminator signal.
> > >
> > > More broadly, we believe the contribution goes beyond this specific instantiation. By bringing imitation learning into FM post-training, we hope to open a useful direction for future work; for example, ideas from inverse reinforcement learning may help inspire more principled reward design in this setting. As Reviewer 8XyN also noted, using a continuous discriminator signal for model alignment is an interesting direction. In addition, our analysis of discriminator design, including the use of a VLM-based discriminator, is also new to the best of our knowledge in this setting.
> > >
> > > We hope this perspective helps clarify the novelty we see in the work and can be taken into account in your final assessment.

---

### Official Review · Reviewer_RJj3 · 2026-03-13

**Soundness:** 3
**Presentation:** 3
**Significance:** 3
**Originality:** 3
**Overall Recommendation:** 3
**Confidence:** 5

**Summary:**

This paper proposes Flow Matching Adversarial Imitation Learning (FAIL), a discriminator-based objective that aligns a student model to an expert image distribution without explicit reward models or preference pairs (Secs. 1–2). The method has two instantiations: FAIL-PD backpropagates through the differentiable ODE solver using a single-step denoising approximation, whereas FAIL-PG treats discriminator scores as rewards and uses an FPO/PPO-style update for black-box or discrete settings (Eq. (2)–(4), Algorithms 1–2). Empirically, the authors fine-tune FLUX.1-dev on 13k single-image-per-prompt demonstrations generated by Gemini Pro 3 and report gains over FLUX, SFT, and RLHF baselines, with results competitive with or better than Online DPO on UniGen, DPG, HPSv3, and UnifiedReward (Table 1). The paper further claims that FAIL-PD is slower but more stable than FAIL-PG over long horizons (Fig. 2), that FAIL regularizes reward-based optimization (Table 3), and that the approach transfers to discrete image and video generation (Tables 4–5, pp. 6–7).

**Compliance With Llm Reviewing Policy:**

Affirmed.

**Final Justification:**

Weak Accept. I appreciate that the authors have resolved many of the raised concerns.

**Key Questions For Authors:**

Please see the Weaknesses [W1]-[W3]. I encourage the authors to address the concerns outlined above in the rebuttal. If these issues are satisfactorily resolved, I would be open to revising my assessment and increasing my final score.

**Limitations:**

yes

**Strengths And Weaknesses:**

[S1]The controlled image-generation results show meaningful gains under a constrained data regime. With only 13k single-image demonstrations, Table 1 reports FAIL-PG at 64.75 UniGen / 84.59 DPG / 11.41 HPSv3 versus SFT at 58.22 / 80.34 / 9.28, and FAIL-PD improves UnifiedReward to 3.4468 at 800 steps (p. 5).

[S2] The paper includes an informative stability analysis rather than only final scores. Sec. 3.3 and Fig. 2 show a concrete speed–stability trade-off: FAIL-PG improves rapidly but collapses after longer training, whereas FAIL-PD improves more slowly.

[S3]At system level, FAIL-PD substantially improves the FLUX.1 Dev backbone: from 83.84 to 87.32 on DPG (Table 6), from 61.30 to 73.70 on UniGen (Table 7), and from 10.43 to 11.28 on HPDv3 (Table 8), all on pp. 7–8.


[W1] Key theoretical steps are asserted rather than validated. FAIL-PD depends on the single-step denoising approximation in Eq. (2) and a local-linearity assumption (p. 2), yet there is no ablation against full-trajectory backpropagation or analysis of the approximation error. Likewise, the “unbiased limit of DDPG” claim and the ELBO/CFM-based KL approximation for FAIL-PG (Appendix A, p. 13) are not rigorously justified.

[W2] The paper’s “preference alignment” evidence is indirect. The target distribution is produced by a synthetic expert generator (called “Nano Banana pro” in the abstract), not by human preference annotations, and the preference metrics are automated proxies (HPSv3 and UnifiedReward) rather than human studies. This supports expert imitation more directly than human alignment.

[W3] The efficiency claim is incomplete. The paper motivates FAIL as avoiding the cost of preference pairs or reward modeling, but the reported setup still requires online rollouts, a trained discriminator, and 32 NVIDIA H20 GPUs with batch size 128 (Sec. 3.1, p. 4). No wall-clock or compute comparison against SFT, DPO, or RLHF is provided.

---

> ### Author Rebuttal · Authors · 2026-03-30
>
> We thank the reviewer for the careful reading and detailed feedback. To save space, we cross-reference results that are also reported in our responses to other reviewers.
>
> > Q1/Q2: Theoretical analysis of one-step denoising approximation and omitting full-trajectory BP.
>
> **These points are the same as Reviewer BqcN’s Q1/Q2**, so we summarize briefly here. Eq. (2) follows directly from the FM linear interpolation path and a one-step Euler update. Empirically results show reconstruction error remains moderate.
>
> We did not use full-trajectory BP because: (1)It is memory-prohibitive for a 12B model with a 28-step ODE solver, because the full computation graph must be retained across all denoising steps. (2)Prior work such as DRaFT [1] shows that BP through long diffusion chains can suffer from noisy or vanishing gradients, while one-step or small-K BP remains competitive and much more efficient.
>
> > Q3: Justification of the KL approximation for FAIL-PG.
>
> Our use of the CFM loss difference is intended as a local proxy for the marginal KL, rather than an exact identity.
>
> From the marginal perspective, following Kingma [2] and FPO, $L_{CFM}$ serves as an ELBO proxy for $log \pi_{\theta}(x)$. Because FAIL-PG uses PPO-style clipped updates together with an explicit KL constraint, the policy is kept close to the reference policy, so the ELBO tightness gap is expected to vary slowly in this local regime. Under this assumption, the difference in CFM losses is a reasonable local surrogate for the marginal log-likelihood ratio.
>
> From the trajectory perspective, prior SDE analyses show that the trajectory-space KL can be expressed in terms of the expected squared $L_2$ distance between vector fields. By the data processing inequality,
>
> $D_{KL}(\pi_{\theta}(x_0) || \pi_{ref}(x_0)) \leq D_{KL}(P_{\theta} || P_{ref}),$
>
> so controlling the policy in vector-field / CFM space provides a principled upper bound on how far the final marginal can drift. We agree that the paper should state this more carefully, and we will revise the wording to emphasize that this is a local, computationally tractable proxy.
>
> > Q4: The claim of “unbiased limit of DDPG”.
>
> We agree that this wording is imprecise. Our intent was to provide an intuition for readers familiar with deterministic policy gradients[3]. A more accurate description is that FAIL-PD is closer in spirit to model-based deterministic policy gradients / SVG[4], where rewards are backpropagated through known differentiable dynamics. We will revise this statement accordingly.
>
> > Q5: The indirectness of the preference-alignment evidence.
>
> Our central claim is about post-training toward a preferred target distribution, which we formulate as imitation learning. In this view, the “expert” distribution is the alignment target. Using a frontier model such as Gemini 3 Pro as the expert gives a practical approximation to a distribution that has already been optimized toward human preferences. If FAIL could minimize divergence with this expert distribution, it inherently capable of aligning with any human-curated distribution.
>
> While we do not use human study, the metrics used are not arbitrary proxies:
> - HPSv3 and UnifiedReward are trained on large-scale, real-world human preference data.
> - We do not use these reward models during training. Performance gains on the out-domain Alchemist dataset confirm that FAIL internalizes general human-preferred aesthetics and semantic coherence rather than merely mimicking expert artifacts.
>
> To address “expert imitation”, we conduct MMD and SWD analysis, which empirically prove that FAIL effectively minimizes the mathematical distance between policy and target distribution. **Please refer to our response to Reviewer 8XyN’s Q2 Tab.6.**
>
> > Q6: Explanation of efficiency and computational overhead.
>
> Our efficiency claim is primarily about supervision cost rather than total wall-clock time: FAIL removes the need for pairwise preference data and a separately trained reward model, and uses only expert demonstrations.
>
> Computationally, compared with other on-policy methods, the main extra cost is the discriminator update. Under our setup, the per-step wall-clock breakdown is:
>
> **Tab.3 Wall-Clock Time Comparison per step**
> | Phase | FAIL-PD (s) | FAIL-PG (s) |
> | :--- | :--- | :--- |
> | Rollout | 39.88 | 40.73 |
> | Discriminator Update | 0.79 | 1.04 |
> | Policy Update | 12.06 | 50.08 |
> | Total Time | 52.73 | 91.85 |
>
> As shown in Tab.3, the discriminator update is small relative to the rollout cost, which is shared by on-policy baselines. In terms of memory, for a 2B discriminator sharded across 32 GPUs with FSDP, the additional optimizer-state overhead is approximately 0.125 GB per GPU.
>
> [1] Directly fine-tuning diffusion models on differentiable rewards
>
> [2] Understanding Diffusion Objectives as the ELBO with Simple Data Augmentation
>
> [3] Deterministic Policy Gradient Algorithms
>
> [4] Learning Continuous Control Policies by Stochastic Value Gradients

---

### Official Review · Reviewer_uo5X · 2026-03-13

**Soundness:** 4
**Presentation:** 3
**Significance:** 3
**Originality:** 3
**Overall Recommendation:** 5
**Confidence:** 3

**Summary:**

The paper introduces Flow Matching Adversarial Imitation Learning (FAIL), a post-training alignment method for visual generative models. Classic alignment typically relies on SFT, which suffers from policy drift, or RL, which requires expensive datasets of human preference pairs or explicit reward models. FAIL bypasses these limitations by framing alignment as an adversarial imitation learning problem. Using a small set of high-quality "expert" images, the method trains a discriminator to distinguish between policy-generated and expert images, while the policy learns to fool the discriminator. The authors propose two optimization variants: FAIL-PD (Pathwise Derivative) and FAIL-PG (Policy Gradient). Evaluated on FLUX.1-dev with 13,000 expert samples, the method demonstrates competitive prompt-following and aesthetic performance compared to DPO and RLHF baselines, and effectively mitigates reward hacking when combined with explicit reward models.

**Compliance With Llm Reviewing Policy:**

Affirmed.

**Key Questions For Authors:**

* Given the severe mode collapse observed in FAIL-PG after ~450 steps, what specific strategies (e.g., KL penalties, early stopping) do you recommend to safely deploy this variant in practice?

**Limitations:**

yes

**Strengths And Weaknesses:**

Strengths:
* Eliminating the reliance on large-scale pairwise preference data removes a significant bottleneck in modern generative AI alignment. Achieving strong results with only 13K single-sample expert demonstrations is highly data-efficient.
* The dual approach is elegant and innovative. FAIL-PD provides a stable, white-box solution leveraging the differentiable nature of flow matching, while FAIL-PG offers a flexible, black-box alternative capable of handling discrete modalities.
* The benchmark improvements (DPG, UniGen, HPSv3) over standard baselines like Online DPO are notable.
* The demonstration of FAIL acting as a dynamic regularizer to prevent reward hacking is a valuable contribution.

Weaknesses:
* The method is validated on a relatively modest dataset size (13K). It is unclear how the adversarial dynamics scale when applied to the massive datasets typically used for frontier model alignment.
* Results show that while FAIL-PG converges rapidly, it suffers from severe mode collapse after approximately 450 steps. This raises concerns about its robustness and hyperparameter sensitivity in generalized deployment.

---

> ### Author Rebuttal · Authors · 2026-03-30
>
> We thank the reviewer for the thoughtful review and helpful feedback. Below we address the main questions.
> > Q1: Data scaling concern.
>
> Because FAIL is an on-policy method, most training samples are generated by the current policy. As in other on-policy methods, training horizon and expert-data quality can matter as much as, or more than, raw dataset size. Fig.2 already suggests that FAIL-PD is the more scalable variant: it continues improving up to 2,000 steps, whereas FAIL-PG saturates much earlier.
>
> Due to time and compute constraints, we could not run experiments beyond 13K expert samples. We did, however, study smaller expert sets (4K and 8K) under the same training budget. As shown in Tab.1, increasing expert data mainly improves prompt-following metrics, while preference metrics change less, which is consistent with the view that preference alignment depends more on data quality than quantity. We also observe that FAIL-PD benefits more clearly from additional data, matching our analysis in Sec. 3.3.
>
> **Tab.1 Ablation of scalable of FAIL**
> | Method | Expert Data Size | Steps | Unigen | DPG | HPSv3 | URScore |
> | :--- | :--- | :--- | :--- | :--- | :--- | :--- |
> | FAIL-PD | 4K | 800 | 60.04 | 83.32 | 11.04 | 3.4379 |
> | FAIL-PD | 8K | 800 | 62.29 | 84.32 | 11.02  | 3.4458 |
> | FAIL-PD | 13K | 800 | 63.38 | 84.55 | 11.17 | 3.4468 |
> | FAIL-PG | 4K | 400 | 63.36 | 84.42 | 11.34 | 3.4433 |
> | FAIL-PG | 8K | 400 | 63.63 | 84.56 | 11.31 | 3.4436 |
> | FAIL-PG | 13K | 400 | 64.75 | 84.59 | 11.41 | 3.4438 |
>
> > Q2: How to stabilize FAIL-PG in extended training.
>
> We attribute the instability of FAIL-PG to two factors: the high variance of policy gradients, and the way KL regularization is used.
>
> Increasing the number of rollouts can reduce gradient variance. For KL regularization, we found that using KL only as an auxiliary loss may be suboptimal. In particular, if the discriminator mistakenly assigns a high reward to a low-quality out-of-distribution sample (a false positive), the policy-loss term can still amplify that sample enough to destabilize training, even if the KL term penalizes it.
>
> A more direct alternative is to incorporate KL into the reward:
>
> $r_{adj}(x) = r_{disc}(x) - \beta \cdot D_{KL}(x)$
>
> This directly lowers the reward of OOD samples during policy optimization.
>
> However, because FAIL-PG uses a dynamic discriminator reward, the reward scale changes during training, which makes $\beta$ difficult to tune. Motivated by the same idea of using KL as a filtering signal rather than only as a loss, we tested a simple KL masking strategy. For each batch, we compute the per-sample KL estimate $KL_i$, define a threshold
>
> $T = mean(KL_i) + c \cdot std(KL_i)$
>
> and mask samples whose KL exceeds $T$ when computing the policy gradient. Intuitively, this removes samples that drift too far from the reference policy and are most likely to be harmful false positives.
>
> With $c=2$, **FAIL-PG remains stable through 800 steps without collapse**. As shown in Tab.2, the longer training horizon gives modest gains, although it still does not surpass FAIL-PD in long-horizon training. We believe this simple masking strategy may also be useful for other diffusion policy-gradient methods, and FAIL-PG would benefit for longer training horizon beyond benchmark metric, as our visulization results in Fig.4 indicate FAIL-PG optimize fine-grained detail such as text redering as the training goes on.
>
> **Tab.2 Ablation of KL masking**
> | Method | Steps | Unigen | DPG | HPSv3 | URScore |
> | :--- | :--- | :--- | :--- | :--- | :--- |
> | FAIL-PG | 400 | 64.75 | 84.59 | 11.41 | 3.4438 |
> | FAIL-PG with KL masking| 800 | 65.49 | 84.81 | 11.45 | 3.4507 |

---

### Decision · Program_Chairs · 2026-04-30

**Decision:**

Accept (regular)

**Comment:**

### Summary
This paper formulates the post-training of flow matching image generation models as an imitation learning problem, where the expert is a high quality dataset. It proposes an approach called flow matching adversarial imitation learning (FAIL) and proposes some concrete algorithms to fine tune flow matching models using this approach. Empirical evaluation with 13000 demonstrations empirically evaluates the proposed techniques.

### Meta-review
The reviewers agree that the proposed technique bridges a known gap in image generation models, and the experimental evaluation validates the hypothesized approaches. There was some concern about scalability and stability of the adversarial procedure but discussion during rebuttal seems to have allayed these issues. I encourage the authors to summarize and include the takeaways from these discussions into the paper.